# CRAM: A COMPRESSION-AWARE MINIMIZER

**Alexandra Peste**[1][*]   **Adrian Vladu**[2]   **Eldar Kurtic**[1]   **Christoph H. Lampert**[1]   **Dan Alistarh**[1,3]
[1]Institute of Science and Technology Austria (ISTA)   [2] CNRS & IRIF   [3] Neural Magic, Inc.

## ABSTRACT

Deep neural networks (DNNs) often have to be compressed, via pruning and/or quantization, before they can be deployed in practical settings. In this work we propose a new compression-aware minimizer dubbed CrAM that modifies the optimization step in a principled way, in order to produce models whose local loss behavior is *stable* under compression operations such as pruning. Thus, dense models trained via CrAM should be compressible post-training, in a single step, without significant accuracy loss. Experimental results on standard benchmarks, such as residual networks for ImageNet classification and BERT models for language modelling, show that CrAM produces dense models that can be more accurate than the standard SGD/Adam-based baselines, but which are stable under weight pruning: specifically, we can prune models in one-shot to 70-80% sparsity with almost no accuracy loss, and to 90% with reasonable ($\sim 1\%$) accuracy loss, which is competitive with gradual compression methods. Additionally, CrAM can produce sparse models which perform well for transfer learning, and it also works for semi-structured 2:4 pruning patterns supported by GPU hardware. The code for reproducing the results is available at: https://github.com/IST-DASLab/CrAM.

## 1 INTRODUCTION

The massive recent progress of deep learning models has been accompanied by an increase in computational costs (Thompson et al., 2020). In turn, this has led to significant interest in *model compression* techniques in order to reduce these costs. For many existing models, compression techniques such as distillation (Hinton et al., 2015), pruning (Hoefler et al., 2021) and quantization (Gholami et al., 2021) can usually reduce the number of parameters or FLOPs of a given model by up to an order of magnitude with relatively little accuracy loss. However, performant compression still usually requires re-training or fine-tuning the model separately for each compression target, provided by the user as a target sparsity and/or quantization level. In turn, this compression process can be cumbersome and error-prone, as it requires additional computation and hyper-parameter tuning for each run.

In this work, we propose *Compression-Aware Minimization (CrAM)*, a method for training neural networks, which results in models that are easily compressible *one-shot*, while still being highly-accurate. Specifically, CrAM enables training a single (dense) model, which can later be compressed to different target levels, with minimal or no recalibration. Such flexibility is desirable, as models can be trained once, and then deployed on multiple devices, with different specifications. Having a single model that can be easily configured to meet the computational requirements of a specific device can both reduce the overall computational cost, and also allow easier customization to individual devices.

CrAM is loosely-inspired by the recently-introduced sharpness-aware minimizer (SAM) (Foret et al., 2021), which trains models that potentially converge to flatter minima, leading to better generalization compared to SGD-type baselines, by biasing the process towards minima of *uniformly low loss*. Multiple subsequent works have investigated and improved upon the original SAM algorithm, by either obtaining better generalization (Kwon et al., 2021), or by reducing the computational costs of SAM training (Liu et al., 2020; Du et al., 2022a). We are the first to carry over this idea to the task of obtaining *compressible* models. Roughly speaking, CrAM works by optimizing not against the original "dense" model, but over a compression projection applied to the intermediate model iterate, at every optimization step. Thus, the CrAM update aims to bias optimization towards iterates that have both *low loss* and are *robust under one-shot compression*. Similarly to SAM, CrAM is simple to implement as part of a regular training loop and has a single scaling hyper-parameter, for which we

---

[*]Correspondence to: alexandra.peste@ist.ac.at

provide a well-performing default value. We detail the CrAM algorithm and provide a theoretical motivation leveraging fundamental results in robust optimization (Danskin, 2012) in Section 3.

To complement our algorithmic contribution, we perform an extensive experimental analysis of CrAM. We mainly focus on compression via weight pruning, but we also show that CrAM is compatible with weight quantization. Generally, CrAM models trained on large-scale image classification or language modelling tasks can improve over the dense baseline performance, while being very robust to one-shot pruning, at different sparsity levels. For image classification, CrAM can train a highly-accurate dense ResNet50 model on ImageNet, that can be pruned *in one-shot* to 80% and 90% sparsity, and is competitive in terms of accuracy relative to state-of-the-art *gradual pruning methods*, following an inexpensive Batch Normalization re-tuning step on a small calibration set.

Moreover, we show that full CrAM training is not necessary for good performance: specifically, a short CrAM finetuning period is sufficient to substantially improve one-shot pruning accuracy. For instance, using CrAM to transfer the standard BERT-base model (Devlin et al., 2019) on SQuADv1.1 question-answering (Rajpurkar et al., 2016), we obtain models that are both more accurate and more compressible than with optimizers such as Adam (Kingma & Ba, 2015) or SAM (Foret et al., 2021). In addition, a short ($\leq 2$ epochs) finetuning of the *sparse* model can provide substantial additional improvements: the 80%-sparse CrAM finetuned model reaches higher accuracy than the highly-competitive gradual pruning methods PLATON (Zhang et al., 2022) and Movement Pruning (Sanh et al., 2020), at a fraction of the training budget.

CrAM lends itself to several extensions: it can be used with different layer-wise sparsity distributions, semi-structured N:M sparsity patterns, and one-shot pruning techniques. Sparse CrAM models can be successfully used for sparse transfer learning, where they can perform well on a wide range of "downstream" target tasks, even when compared to pruning methods that train a separate model for each sparsity level. We also provide evidence that the CrAM update can produce models that are robust to quantization.

Similar to SAM (Foret et al., 2021), one limitation is the added computational cost, as CrAM requires an additional backwards pass for the model perturbation. This can be addressed by only performing limited finetuning via CrAM instead of full retraining, or by only performing a regular optimization step for a fraction of the time, both of which we show to have a limited impact on accuracy. Moreover, our approach is also compatible with efficient SAM-type updates (Liu et al., 2020; Du et al., 2022a). We also provide a well-performing variant of CrAM that uses sparse gradients, which could be leveraged by frameworks with support for sparse back-propagation (Nikdan et al., 2023).

## 2    RELATED WORK

**Sharpness-Aware Minimization (SAM).** The recently introduced SAM optimizer (Foret et al., 2021) aims to minimize loss *sharpness*; this in turn should lead to flatter local minima, with better generalization. The authors show that SAM-trained models have higher validation accuracy compared to vanilla SGD-type baselines, that their performance continues to improve with prolonged training, and that they can also be successfully used for transfer learning. One important drawback of SAM is its computational overhead, as it requires twice as many forward-backward passes through the network. Subsequent work has focused on reducing computational cost by, for example, reducing the frequency of the extra gradient steps (Liu et al., 2022), computing the perturbations on a subset of the parameters (Du et al., 2022a), or by proposing a new trajectory loss to replace the sharpness definition (Du et al., 2022b). We draw inspiration from properties of the initial SAM method proposed by Foret et al. (2021). Instead of attempting to minimize the maximum local increase loss (sharpness), our goal is to minimize the maximum local increase in loss due to compression.

**Training prunable networks.** The increasing scale of deep neural networks have made their deployment to edge devices dependent on compression techniques, such as quantization and/or pruning. While post-training quantization can be efficient and successful without any retraining (Frantar & Alistarh, 2022), in the case of pruning the gold standard is still training a separate model for every target sparsity level (Zhu & Gupta, 2017; Singh & Alistarh, 2020; Evci et al., 2020; Peste et al., 2021), which can be expensive. A potential solution would be training a single dense model, which either contains multiple smaller ones that can be easily deployed, or which is itself *prunable* at multiple sparsity levels, without additional retraining. For example, "once-for-all" (OFA) (Cai et al., 2019) can train a large network that contains multiple specialized sub-nets, adapted to different resource constraint devices. However, obtaining the large OFA network is extremely expensive, and requires

intensive finetuning to ensure a good performance for the sub-nets. A similar idea has also been explored for automatic speech recognition (Wu et al., 2021). An orthogonal direction is to obtain "slimmable neural networks" (Yu et al., 2019; Yu & Huang, 2019b;a), by training a single model that can be executed at different widths; this is usually achieved by performing multiple backpropagations using all the predefined widths, at each optimization step. Furthermore, BigNAS (Yu et al., 2020) is inspired by these approaches, and achieves a similar goal to OFA, without additional finetuning. Compared to our method, these works focus on structured pruning, and require extensive finetuning, or more forward-backward steps at each iteration. Related to one-shot pruning, Only Train Once (OTO) (Chen et al., 2021b) has been proposed for structured pruning, to train a large model that is easily slimmable one-shot. While we obtain better results than OTO for the same sparsity level, the two methods are not directly comparable, since we focus on *unstructured sparsity*.

Our work is more closely related to Miao et al. (2022); Zimmer et al. (2022), which propose leveraging Stochastic Frank-Wolfe (SFW) (Reddi et al., 2016) to encourage the weights to lie in a convex hull spanned by sparse vectors; this would make the model prunable one-shot, without any finetuning. The methods proposed in Miao et al. (2022); Zimmer et al. (2022) result in highly-prunable models on relatively-small tasks. Under the same experimental setting, CrAM is able to match or outperform these methods; for instance, CrAM can prune VGG-16 trained on CIFAR-10 in one-shot to 95% sparsity without accuracy loss, outperforming SFW by more than 2% Top-1 accuracy. More importantly, we show that CrAM produces models compressible in one-shot at both ImageNet scale and BERT language modeling scale. Remarkably, with one-shot pruning CrAM can offer competitive performance to *gradual pruning* methods, whether they are designed for CNNs (Kusupati et al., 2020; Lin et al., 2020) or for language modelling (Sanh et al., 2020; Zhang et al., 2022).

## 3 THE COMPRESSION-AWARE MINIMIZER (CRAM)

### 3.1 BACKGROUND

We now give an overview of our method, together with the corresponding algorithm and generalizations. CrAM aims to train models that are "compressible" in one-shot, following training, via sparsity or quantization. For this, we consider a compression operator $C$, such as for example Top-K, which keeps the highest $K$ elements in absolute value of a tensor, and sets the rest to 0. We say that a model is easily compressible if small perturbations do not affect its performance after compression. To enforce this during training, we optimize against the dense model perturbation which has the largest impact on the compressed model. We want to minimize the "compression-aware" (CrAM) loss, defined as:

$$L^{\text{CrAM}}(\boldsymbol{\theta}) = \max_{\|\boldsymbol{\delta}\| \leq \rho} L(C(\boldsymbol{\theta} + \boldsymbol{\delta})), \tag{1}$$

where $\boldsymbol{\theta}$ is the vector of parameters, $L$ is the model loss and $\boldsymbol{\delta}$ is a norm-bounded perturbation.

We approximate $L^{\text{CrAM}}(\boldsymbol{\theta})$ by taking a gradient ascent step in the direction of the current update, followed by a projection using the compression operator. This is inspired by the iterative hard thresholding (IHT) algorithm used for optimizing functions under sparse constraints (Blumensath & Davies, 2008; Foucart, 2011; 2012). To obtain the gradient with respect to the parameters, we employ a straight-through estimator, and use the gradient evaluated at the perturbation. This gives us the following update for minimizing the CrAM loss at step $t + 1$:

$$\widetilde{\boldsymbol{\theta}}_t = C(\boldsymbol{\theta}_t + \rho \cdot \nabla L(\boldsymbol{\theta}_t)) \qquad \boldsymbol{\theta}_{t+1} = \boldsymbol{\theta}_t - \eta \nabla L(\widetilde{\boldsymbol{\theta}}_t). \tag{2}$$

Alongside improving robustness to compression, we want to maintain the quality of the dense model, which cannot be guaranteed by only optimizing the CrAM loss. Therefore, we propose to explicitly optimize for the performance of the dense model. The main objective we use and which fulfills these criteria is the composite *CrAM$^+$ loss function*, defined as:

$$L^{\text{CrAM}^+}(\boldsymbol{\theta}) = L(\boldsymbol{\theta}) + L^{\text{CrAM}}(\boldsymbol{\theta}) \tag{3}$$

This requires a simple modification to the CrAM update, at no extra cost, by simply adding the gradient $\nabla L(\boldsymbol{\theta}_t)$, before the next update of the parameters $\boldsymbol{\theta}_{t+1}$. For $\widetilde{\boldsymbol{\theta}}_t = C(\boldsymbol{\theta}_t + \rho \nabla L(\boldsymbol{\theta}_t))$, the CrAM$^+$ update is the following:

$$\boldsymbol{\theta}_{t+1} = \boldsymbol{\theta}_t - \eta \cdot (\nabla L(\widetilde{\boldsymbol{\theta}}_t) + \nabla L(\boldsymbol{\theta}_t)). \tag{4}$$

We can also add different regularization terms to the objective in Equation 3, such as weight decay.

---

**Algorithm 1** Compression-Aware Minimization (CrAM / CrAM$^+$)

---

**Require:** Compression operators $\mathcal{C} = \{C_1, C_2, \dots, C_M\}$, training data $S$, training iterations $T$, learning rate $\eta$, perturbation step size $\rho$
1: Initialize the weights $\boldsymbol{\theta}_0$
2: **while** $t \leq T$ **do**
3:    Sample batch $x \in S$
4:    Compute loss $L(\boldsymbol{\theta}_t; x)$ and gradient $\boldsymbol{g}_t = \nabla L(\boldsymbol{\theta}_t; x)$
5:    Uniformly choose a compression operator $C \in \mathcal{C}$
6:    Get perturbed weights $\widetilde{\boldsymbol{\theta}}_t = C(\boldsymbol{\theta}_t + \rho \boldsymbol{g}_t)$
7:    **if** $C$ = Top-K **then**
8:       Let $M_t$ be the linear projection operator onto the support of the largest $K$ coordinates of $|\boldsymbol{\theta}_t + \rho \boldsymbol{g}_t|$, such that $\widetilde{\boldsymbol{\theta}}_t = M_t(\boldsymbol{\theta}_t + \rho \boldsymbol{g}_t)$
9:       $\widetilde{\boldsymbol{g}}_t = M_t \nabla L(\widetilde{\boldsymbol{\theta}}_t; x)$
10:    **else**
11:       $\widetilde{\boldsymbol{g}}_t = \nabla L(\widetilde{\boldsymbol{\theta}}_t; x)$
12:    **end if**
13:    **if** use CrAM$^+$ **then**
14:       $\widetilde{\boldsymbol{g}}_t \leftarrow \widetilde{\boldsymbol{g}}_t + \boldsymbol{g}_t$
15:    **end if**
16:    Update the weights using a gradient descent step: $\boldsymbol{\theta}_{t+1} = \boldsymbol{\theta}_t - \eta \cdot \widetilde{\boldsymbol{g}}_t$
17: **end while**
18: **return** $\boldsymbol{\theta}_T$

---

Furthermore, we can consider a distribution over a set of compression operators $\mathcal{C}$ and approximate the expected CrAM loss under multiple types of compression, by sampling a different operator at each step. We refer to this version as CrAM/CrAM$^+$-Multi.

## 3.2 THEORETICAL JUSTIFICATION OF THE CRAM UPDATE

To derive the CrAM update, and justify the choices made in designing our training method, we start from the optimization objective defined in Equation 1. Although the loss includes an inner maximization problem, together with a potentially problematic compression operator, we show that under mild assumptions we can efficiently approximate a descent direction. We rely on a well-known theorem from robust optimization (Danskin, 2012), which allows one to obtain descent directions for min-max objectives under a broad range of assumptions. Using Danskin's theorem (Theorem 1 from Appendix A.1) we obtain that by computing the maximizer of the inner problem

$$\boldsymbol{\delta}^* = \arg \max_{\|\boldsymbol{\delta}\| \leq \rho} L(C(\boldsymbol{\theta} + \boldsymbol{\delta})), \tag{5}$$

and letting $\boldsymbol{\phi} = \boldsymbol{\theta} + \boldsymbol{\delta}^*$, which compresses to the extrapolated iterate $\widetilde{\boldsymbol{\theta}} = C(\boldsymbol{\phi})$, we obtain a descent direction $-\nabla L(C(\boldsymbol{\phi}))$. Implementing this approach faces two difficulties – first, to compute the gradient we must back-propagate through the composition of functions $L(C(\cdot))$, which may cause trouble since $C$ is not necessarily differentiable; second, and more importantly, it is unclear how to solve the inner maximization problem.

To address the first issue, we may choose to use a straight-through gradient estimator (Bengio et al., 2013), which permits us to only backpropagate through $L$, and use $\nabla L(\widetilde{\boldsymbol{\theta}})$ instead of the true gradient. To increase precision, in the case where compression is performed via Top-K we interpret $C$ as a "mask" operator $M$ which zeroes out a subset of coordinates dependent on $\boldsymbol{\phi}$. Since except for articulation points, $M_t$ is constant and does not change as the argument varies, we approximate $\nabla L(C(\boldsymbol{\phi})) \approx M \nabla L(M\boldsymbol{\phi}) = M \nabla L(\widetilde{\boldsymbol{\theta}})$.

To address the second issue, rather than exactly maximizing the inner problem, we instead seek a good enough maximizer using a standard iterative method. For this, we choose projected gradient ascent, which provides theoretical guarantees, even when the projection is performed onto non-convex domains (Peste et al., 2021). For instance, if the compression operator is magnitude pruning, this becomes the iterative hard thresholding (IHT) method, frequently employed in the sparse recovery literature (Blumensath & Davies, 2008). Thus, to reach a good iterate within this specific domain, in practice we perform a single step of (projected) gradient ascent, which matches the IHT iteration:

$$\widetilde{\boldsymbol{\theta}}_t = C(\boldsymbol{\theta}_t + \rho \cdot \nabla L(\boldsymbol{\theta}_t)). \tag{6}$$

In Appendix A, we provide a full re-derivation of the CrAM update in Equation 2 under fairly reasonable assumptions on the objective function, along with a detailed discussion on the necessity of these assumptions; our derivations also hold for CrAM$^+$. As a side result, we also obtain a simple re-derivation of the SAM update.

### 3.3 IMPLEMENTATION DETAILS AND EXTENSIONS

**Multiple compression types.** CrAM can be used to train models that are robust to multiple types of compression operators, by choosing between multiple compression projections at each optimization step. Examples include pruning using different sparsity levels or quantizing at different precisions. We illustrate the general CrAM algorithm which handles multiple compression operators, and includes the explicit optimization of the dense model, in Algorithm 1. We found that CrAM$^+$ with a different randomly chosen compression at each optimization step typically achieves the best trade-off between an accurate dense model and robustness to multiple one-shot compression schemes post-training; we call this variant *CrAM$^+$-Multi*, and will become the main method used in our experiments. When optimizing for robustness against sparse perturbations, we use the Top-K operator at each step, and choose the sparsity level uniformly at random among a set of predefined values.

**Addressing the computational overhead.** While CrAM requires twice as many forward-backward passes compared to regular training, we can reduce this overhead for TopK-CrAM, by making use of the sparsity in the intermediate updates. We found that using only the gradients from the support of $\widetilde{\boldsymbol{\theta}}_t$ in $\nabla L(\widetilde{\boldsymbol{\theta}}_t)$ improves both the resulting dense model obtained with TopK-CrAM, as well as its robustness to one-shot pruning. This observation is motivated by the fact that the straight-through (Bengio et al., 2013) gradient estimator with the identity function might be suboptimal (Yin et al., 2019), and better estimators can be defined using different functions. As seen in Section 3.2, we can assume, via Danskin's theorem (Danskin, 2012), that we can obtain descent directions for $L^{\mathrm{CrAM}}(\boldsymbol{\theta}_t)$ by evaluating $\nabla L(C(\boldsymbol{\phi}_t))$, where $\boldsymbol{\phi}_t$ is the extrapolated point $\boldsymbol{\phi}_t = \boldsymbol{\theta}_t + \rho \nabla L(\boldsymbol{\theta}_t)$. To evaluate the gradient, we may use a straight-through estimator. For Top-K, $C$ is as an operator $M_t$ which zeroes out a subset of coordinates dependent on $\boldsymbol{\phi}_t$. Provided that $M_t$ is constant and does not change as the argument varies, we can approximate $\nabla L(C(\boldsymbol{\phi}_t)) \sim M_t \odot \nabla L(M_t \boldsymbol{\phi}_t)$. As both the iterate and gradient estimator are sparse, this implies a theoretical speed-up. We further show in Appendix C.6 results obtained from CrAM using infrequent mask updates.

**Alternative Updates.** We note that alternative compression-aware updates can be derived. One such derivation, which we call Compressed-SAM (C-SAM), uses the gradient of the compressed model to perturb the dense model. While training with C-SAM can also result in models robust to one-shot pruning, we noticed that typically the accuracy of the resulting dense models is lower, compared to using CrAM. Moreover, CrAM can be easily modified to also optimize the dense model, while for C-SAM this modification would require an additional forward-backward pass. We also examine the importance of the extra gradient step in CrAM, by comparing against simply applying the Top-K operator to the parameters. We provide an ablation study in Appendix C.1.

**Statistics Correction.** It is well-known (Hubara et al., 2021; Frantar & Alistarh, 2022) that pruning weights in a single step at high sparsity levels can have a large negative effect on normalization layers, due to a mismatch between layer statistics, e.g. the running mean and variance of Batch Normalization (BN) layers, computed during training, and those of the pruned model. To correct for this, following prior work, we keep a subset of randomly chosen 1000 training samples, to which we apply standard training augmentations, and which are used post-pruning for resetting the BN statistics of the pruned model. We note that this procedure, which we refer to as Batch Norm Tuning (BNT) is very inexpensive, and does not finetune any model parameters. Furthermore, when training CrAM training for image classification, we only track the BN statistics on the dense model, before applying the compression perturbation. In the case of BERT models, we do not apply any statistics corrections.

## 4 EXPERIMENTS

Our experimental validation mainly focuses on sparsity, obtained by applying the Top-K operator, in the context of CrAM (i.e. TopK-CrAM). The main method we propose is CrAM$^+$ with multiple sparsity levels chosen uniformly at random at each step (CrAM$^+$-Multi). We also experiment with particular cases of this method, where only one sparsity level is used (e.g. CrAM$^+$-k70), and also with the initial CrAM method with low sparsity (e.g. CrAM-k50). For image classification experiments,

all one-shot pruning results are presented after BNT on a subset of 1000 training samples, i.e. 100 inference steps on batches of size 128, using standard random augmentations.

## 4.1 IMAGENET EXPERIMENTS

**General Setup.** We use a standard setup for training our ImageNet/ResNet50 models, similar to Foret et al. (2021), which we describe in Appendix B. To match the number of backpropagation steps of CrAM, we additionally train the dense baseline for twice as many epochs. We have found that $\rho = 0.05$ recommended by the authors of SAM (Foret et al., 2021) is a good value for CrAM, and we have kept it for all our ImageNet experiments. As stated, after one-shot pruning, we perform BNT on a subset of 1000 training samples (e.g. one per class), with standard augmentations. We show in the Appendix C.3 that the accuracy after BNT is extremely stable, w.r.t. the choice of calibration set.

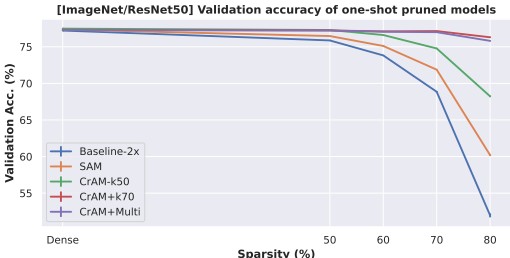

**Figure 1:** One-shot pruning results, averaged over 10 BNT trials using randomly chosen calibration sets of 1000 samples.

**Table 1:** Accuracy of one-shot pruned CrAM⁺-Multi models vs. pruning methods.

| Model | Sparsity 80% | Sparsity 90% |
|---|---|---|
| CrAM⁺-Multi | 75.8 | 74.7 |
| WoodFisher | 76.7 | 75.3 |
| AC/DC | 76.2 | 75.2 |
| STR | 76.1 | 74.3 |
| DPF | 75.1 | 74.6 |

**Results for one-shot pruning.** We validate that different versions of CrAM models are robust to post-training compression, by testing their accuracy after one-shot pruning, at different sparsity levels. We train models using CrAM-k50, CrAM⁺-k70 and CrAM⁺-Multi and global magnitude; for CrAM⁺-Multi we choose the sparsity level randomly at each step from the set $\{50\%, 70\%, 90\%\}$. Using CrAM⁺ is crucial for preserving the dense model accuracy, at higher sparsities ($\geq 70\%$) during training; however, when training with low sparsities, vanilla CrAM can result in dense models that are slightly better than the baseline (e.g. CrAM-k50). For both CrAM⁺-k70 and CrAM⁺-Multi we use sparse gradients for the Top-K model perturbation, as described in Section 3.2. This improved substantially the accuracy after one-shot pruning, as well as the resulting dense model. Additionally, this could offer training-time speed-up compared to, for example, using dense gradients or training with SAM with the right framework support. We include an ablation study on the effects of sparse gradients in Appendix C.2.

The results from Figure 1 show that CrAM models are substantially more robust to one-shot pruning, compared to standard SGD or SAM training. CrAM models do not lose accuracy at lower sparsity (e.g. at 50% for all or 70% for CrAM⁺-k70 and CrAM⁺-Multi) and the resulting dense models tend to outperform the baseline (e.g. the ImageNet validation accuracy for CrAM⁺-Multi is 77.3% accuracy vs. 76.95% for the baseline). Overall, CrAM⁺-Multi is the most robust to one-shot pruning across different sparsity levels. For example, as shown in Table 1, the results at 80% and 90% sparsity are competitive with state-of-the-art pruning methods, such as WoodFisher (Singh & Alistarh, 2020) or AC/DC (Peste et al., 2021), and even surpass some pruning methods (STR (Kusupati et al., 2020), DPF (Lin et al., 2020)). We emphasize that CrAM requires a single round of training (albeit with twice as many forward-backward passes, compared to regular SGD), while standard pruning methods require training separately for each target sparsity, sometimes from a pretrained model (e.g. WoodFisher). In addition to global magnitude, CrAM can be used successfully with uniform magnitude pruning; we show additional results in the Appendix C.4, as well as evidence that CrAM models are robust to sparse distributions different from those used during training.

**Results for semi-structured sparsity.** We show the robustness of CrAM on semi-structured N:M sparsity patterns, which can provide practical speed-ups (Mishra et al., 2021). CrAM⁺ trained using N:M sparsity preserves the dense model's accuracy (77.3%), and can be accurately pruned one-shot (+BNT) to 2:4 (77.0% ) or 4:8 (77.2%) patterns. This is competitive with state-of-the-art methods for N:M sparsity Zhou et al. (2021). We provide a full discussion in Appendix C.5.

**Finetuning with CrAM.** To reduce the training cost of CrAM, we investigate whether a model can become more robust to pruning after short finetuning with CrAM. We are inspired by Andriushchenko & Flammarion (2022), who showed that similar benefits to full SAM training can be obtained when SAM

**Figure 2:** Average relative increase in error, relative to dense, on 12 tasks, between models pruned one-shot, or obtained from pruning methods, at different sparsities. Lower is better. All models were pretrained on ImageNet/ResNet50. For better visibility, error bars indicate 70% confidence intervals.

is used only in the final training phase. We finetune pretrained ImageNet ResNet18 and ResNet50 models, from the Torchvision library, for 10 epochs, using CrAM$^+$-k70 and CrAM$^+$-Multi, both with sparse gradients. For CrAM$^+$-Multi we randomly select at each step a sparsity level in the range 50%-90%. For comparison, we also finetuned using SGD with momentum or using SAM, under the same hyperparameters. We report in Tables 2 and 3 the validation accuracy for the dense models, and after one-shot pruning at 50%-80% sparsity levels. Finetuning with CrAM$^+$ preserves or outperforms the baseline accuracy, and results in good sparse models after one-shot pruning, at moderate sparsity levels (up to 70%). Results improve with longer finetuning: after 20 epochs, the CrAM$^+$-Multi model can be pruned one-shot to 70% sparsity, with $\leq 1\%$ drop in accuracy, compared to the baseline.

| Model | Dense | Sparsity | | | |
| --- | --- | --- | --- | --- | --- |
| | | 50% | 60% | 70% | 80% |
| SGD | 70.4 | 68.9 | 67.0 | 62.3 | 50.1 |
| SAM | 70.5 | 69.2 | 67.4 | 63.4 | 52.2 |
| CrAM$^+$-k70 | 70.3 | 69.5 | 68.8 | **69.0** | 65.0 |
| CrAM$^+$-Multi | 70.4 | 69.7 | 69.2 | 68.3 | 66.7 |
| CrAM$^+$-Multi-20 | **70.6** | **69.9** | **69.6** | 69.0 | **67.6** |

**Table 2:** (ResNet18) Accuracy after finetuning for dense models, and after one shot pruning (+BNT)

| Model | Dense | Sparsity | | | |
| --- | --- | --- | --- | --- | --- |
| | | 50% | 60% | 70% | 80% |
| SGD | 76.8 | 75.4 | 73.6 | 69.0 | 53.1 |
| SAM | **76.9** | 75.8 | 74.3 | 70.5 | 57.8 |
| CrAM$^+$-k70 | 76.8 | 75.9 | 75.5 | 75.4 | 72.0 |
| CrAM$^+$-Multi | 76.7 | 75.9 | 75.6 | 75.0 | 73.5 |
| CrAM$^+$-Multi-20 | 76.8 | **76.1** | **75.7** | **75.5** | **74.4** |

**Table 3:** (ResNet50) Accuracy after finetuning for the dense models, and after one shot pruning (+BNT)

**Sparse Transfer Experiments.** We additionally test how well the sparse models obtained through one-shot pruning after CrAM training on ImageNet transfer across different tasks. The setup is very similar to Salman et al. (2020); Kornblith et al. (2019), where transfer is performed across 12 benchmark tasks. Following Iofinova et al. (2022), we use full finetuning of the non-zero weights, with fixed masks, and reinitialize the dense classification layer. We compare dense and one-shot pruned CrAM-k50 and CrAM$^+$-Multi models (before BNT), trained on ImageNet/ResNet50 to the corresponding ones obtained from SGD or SAM. In addition, we compare with standard pruning methods used in the literature, such as lottery-tickets (LTH) (Chen et al., 2021a), AC/DC (Peste et al., 2021), STR (Kusupati et al., 2020) or WoodFisher (Singh & Alistarh, 2020), for which we use public finetuned models on the downstream tasks (Iofinova et al., 2022). For each model and sparsity level, we aggregate the results over all tasks, measuring the relative increase in error of the sparse models, compared to the dense baseline, as was done in (Iofinova et al., 2022). The results in Figure 2 show that one-shot pruned CrAM models transfer well. In fact, both CrAM-k50 and CrAM$^+$-Multi models transfer better than LTH or STR at 80% sparsity. Also, CrAM$^+$-Multi at 90% sparsity has a similar transfer performance to AC/DC models, and gives better results compared to the other pruning methods used for comparison (LTH or STR), with the exception of the second-order WoodFisher method, which is the best performing method across both 80% and 90% sparsity. Moreover, the dense CrAM$^+$-Multi model has a very similar transfer performance to the baseline, with a less than 1% average relative increase in error, while dense CrAM-k50 or SAM, as well as the baseline trained for twice more iterations result in a minor improvement in transfer accuracy of around 2%, compared to the baseline. Compared to the standard pruning methods used for comparison, CrAM has the added advantage that it produces an accurate dense model, and both 80% and 90% models with a single ImageNet run.

**Quantization.** We additionally provide evidence that CrAM can be adapted to quantization. Namely, a short finetuning using a quantization version of CrAM on pretrained ImageNet models can preserve

their accuracy with respect to the baseline, after symmetric per-channel 4 bits quantization, while also boosting the accuracy of the dense model. More details can be found in Appendix C.7.

## 4.2 EXPERIMENTS ON LANGUAGE MODELLING

In addition to image classification, we also successfully apply CrAM for language models. We demonstrate that CrAM produces models that are more compressible and accurate than the ones obtained with standard optimizers like Adam (Kingma & Ba, 2015) and SAM (Foret et al., 2021). Also, we show that CrAM models are even competitive with gradual pruning methods, which usually require a higher computational budget to produce accurate sparse models for each sparsity target independently.

**General setup.** We focus on the standard benchmark for compression methods: the BERT-base (Devlin et al., 2019) model on the span-based question-answering task SQuADv1.1 (Rajpurkar et al., 2016). We consider the short fine-tuning setup (1-3 epochs) of the pretrained BERT-base model on a downstream task. Following the community standards (e.g. Sanh et al. (2020), Kurtic et al. (2022)) we sparsify the weights of the encoder part, and make use of the Top-K operator at each step to impose uniform sparsity distribution over all layers.

**Robustness to one-shot pruning.** We fine-tune the model with Adam, SAM, and several variants of CrAM and test their robustness to one-shot pruning with the standard magnitude pruner. To identify the optimal set of hyper-parameters we run a grid search (please see Appendix D for more details) and pick the one with the best one-shot performance at 50% sparsity target. For a fair comparison, we allow Adam to fine-tune for twice as many epochs as SAM and CrAM. The results presented in Table 4 suggest that CrAM models are more robust to one-shot pruning while still being able to match or even outperform the dense accuracy obtained with other optimizers.

| Model | Dense | Sparsity | | | |
|---|---|---|---|---|---|
| | | 50% | 60% | 70% | 80% |
| Adam | 88.7 | 80.0 | 32.5 | 9.6 | 8.1 |
| SAM | 88.5 | 81.0 | 33.4 | 10.1 | 7.3 |
| CrAM$^+$-k50 | **88.9** | **88.3** | 84.6 | 25.3 | 8.3 |
| CrAM$^+$-k60 | 88.7 | 88.1 | 87.8 | 75.7 | 10.2 |
| CrAM$^+$-k70 | 88.8 | 87.8 | 87.0 | **86.9** | 33.9 |
| CrAM$^+$-k80 | 88.4 | 86.9 | 85.5 | 84.9 | **84.7** |
| CrAM$^+$-Multi | 88.7 | **88.3** | **88.1** | 86.8 | 82.5 |

**Table 4:** (SQuADv1.1/BERT-base) Validation F1 score of models after fine-tuning with the corresponding optimizer and applying one-shot magnitude pruning.

| Model | Pruning | Sparsity | | | |
|---|---|---|---|---|---|
| | | 50% | 60% | 70% | 80% |
| $\ell_0$ regularization | gradual | 84.6 | 83.9 | 82.8 | 81.9 |
| Magnitude | gradual | 87.0 | 86.7 | 86.5 | 84.8 |
| Movement | gradual | 83.0 | 82.8 | 81.9 | 82.0 |
| Soft-Movement | gradual | 85.8 | N.A. | 84.6 | N.A. |
| PLATON | gradual | 87.2 | 86.9 | 86.7 | 86.1 |
| CrAM$^+$-Multi | one-shot magnitude | 88.3 | 88.1 | 86.8 | 82.5 |
| | one-shot oBERT | **88.7** | 88.1 | 87.5 | 84.9 |
| | one-shot oBERT + fine-tune | **88.7** | **88.4** | **88.1** | **87.4** |

**Table 5:** (SQuADv1.1/BERT-base) Validation F1 score of the CrAM$^+$-Multi model after one-shot pruning with magnitude and oBERT pruners. We additionally fine-tune the one-shot oBERT-pruned model and compare it with gradual pruning methods.

**Comparison with gradual pruning methods.** We investigate whether CrAM models can be competitive with those produced by gradual pruning methods, which progressively prune and fine-tune the model for many epochs. We adopt the CrAM$^+$-Multi model from Table 4 and prune it in one-shot with the standard magnitude pruner, but also with the state-of-the-art BERT-pruning method called oBERT (Kurtic et al., 2022). Since one-shot pruning to high sparsity targets can severely impact the model's performance, we also investigate whether short fine-tuning (for at most 2 epochs) on top of it can bridge the gap towards full accuracy recovery. In Table 5 we present the results and compare against the following gradual pruning methods: $\ell_0$ regularization (Louizos et al., 2018), Magnitude (Zhu & Gupta, 2017), Movement (Sanh et al., 2020), Soft-Movement (Sanh et al., 2020) and PLATON (Zhang et al., 2022). For details regarding hyper-parameters, please see Appendix D. As can be seen from the results, one-shot pruned CrAM models are competitive with gradual pruning methods, which they outperform by huge margins when additionally fine-tuned for a few epochs. It is worth emphasizing that competitive results obtained with two different one-shot pruners, magnitude and oBERT, suggest that CrAM models are indeed robust and compatible with different pruning techniques than the ones they have been trained with. We provide in Appendix D Tables 20 and 19 speed-ups for the sparse BERT models, and evidence that models become more robust to pruning even when CrAM is not used at every optimization step.

## 4.3 DETAILED COMPARISONS WITH OTHER METHODS

We perform a detailed comparison with other methods on CIFAR-10 (Krizhevsky et al., 2009), which is a common setup for all previous methods we consider. Specifically, we compare CrAM with

state-of-the-art gradual pruning methods (Lin et al., 2020) on ResNet20 (He et al., 2016), or with similar methods that train prunable networks (Miao et al., 2022) on VGG-16 (Simonyan & Zisserman, 2014) or ResNet18 (He et al., 2016). We discuss all hyperparameters in Appendix B, and compare against one-shot pruning from dense baselines in Appendix C.8.

**Comparison with Gradual Methods.** We train CrAM$^+$-Multi, with sparse intermediate gradients, and sparsity values chosen randomly among $\{50\%, 70\%, 80\%, 90\%, 95\%\}$. The results in Table 6 show that the *dense* CrAM$^+$-Multi model is highly accurate, and also very robust to one-shot pruning, even at high sparsities (e.g. 90% and 95%). Remarkably, our results for one-shot pruning (+BNT) are usually competitive with those obtained by other methods which train sparse models separately, for each sparsity target, for example through DPF (Lin et al., 2020). Notably, we substantially outperform DPF at 95% sparsity, after BNT.

| Architecture | Model | Dense | Sparsity | | | | |
| --- | --- | --- | --- | --- | --- | --- | --- |
| | | | 50% | 70% | 80% | 90% | 95% |
| ResNet20 | CrAM$^+$-Multi | $92.9 \pm 0.$ | $92.8 \pm 0.1$ | $\mathbf{92.7 \pm 0.2}$ | $\mathbf{92.6 \pm 0.2}$ | $\mathbf{91.2 \pm 0.1}$ | $\mathbf{89.2 \pm 0.1}$ |
| | DPF | N/A | N/A | $92.4 \pm 0.1$ | $92.2 \pm 0.2$ | $90.9 \pm 0.1$ | $88.0 \pm 0.3$ |
| VGG-16 | CrAM$^+$-k95 | $\mathbf{94.2 \pm 0.1}$ | $\mathbf{94.2 \pm 0.1}$ | $\mathbf{94.2 \pm 0.1}$ | $\mathbf{94.1 \pm 0.1}$ | $\mathbf{94.0 \pm 0.1}$ | $\mathbf{94.1 \pm 0.1}$ |
| | SFW | N/A | 93.1 | 93.1 | 93.1 | 93.1 | 92.0 |
| | DPF | N/A | N/A | N/A | N/A | N/A | $93.9 \pm 0.2$ |

**Table 6:** (CIFAR-10) Test accuracy (%) for CrAM after one shot pruning (+BNT). CrAM$^+$-Multi can outperform gradual pruning method DPF, up to 95% sparsity. DPF requires retraining for each target sparsity. CrAM$^+$ outperforms similar method SFW (Miao et al., 2022).

**Comparison with One-shot Methods.** We compare against other methods for training prunable models, such as Miao et al. (2022); Zimmer et al. (2022). Both are based on Stochastic Frank-Wolfe (SFW) (Reddi et al., 2016), to encourage the parameters to lie in the convex hull spanned by sparse vectors, with directions given by the gradients. We compare CrAM against SFW for CIFAR-10, on ResNet18 (He et al., 2016) and VGG-16 models, which is the same setup used in Miao et al. (2022) and Zimmer et al. (2022). We train CrAM$^+$-k95 with sparse intermediate gradients, for 180 epochs (same as Miao et al. (2022)), using SGD with momentum and weight decay, and a cosine learning rate schedule. We use BNT after pruning. On both ResNet18 and VGG-16, we obtain dense models that preserve the baseline accuracy: 94.2% (CrAM$^+$-k95) vs. 93.9% (dense) on VGG-16 and 95.7% (CrAM$^+$-k95) vs. 95.4% (dense) on ResNet18. Furthermore, on ResNet18 we maintain the model accuracy after pruning one-shot at 96% sparsity (95.4%, after BNT) and have a 1.3% drop at 98% sparsity (94.4% Top-1); in comparison, Miao et al. (2022) and Zimmer et al. (2022) obtain $\leq 93\%$ accuracy at 95% sparsity. We show a numerical comparison for VGG-16 in Table 6: CrAM$^+$-k95 preserves model accuracy even at 95% sparsity, which is competitive with the DPF gradual method, while SFW produces models that have lower accuracy even at higher density. CrAM has higher training costs than SFW, but requires much less hyper-parameter tuning, and leads to higher accuracies. Similar to Zimmer et al. (2022), we use BNT, but the additional cost of BNT is minimal; even *without BNT*, we preserve accuracy at up to 80% sparsity (Appendix C.8), leading to better results than Miao et al. (2022); Zimmer et al. (2022).

## 5 CONCLUSIONS AND FUTURE WORK

We proposed a new method for training neural networks, CrAM, which results in models that are both highly accurate, and easily-compressible. Our extensive experimental analysis on large scale image classification (ImageNet/ResNets) and language modelling (SQuADv1.1/BERT-base) shows that CrAM models can be pruned one-shot at a wide range of sparsity levels, while resulting in sparse models that are competitive with existing gradual pruning methods. Furthermore, we show that one-shot pruned CrAM models can transfer better to downstream tasks, compared to some of the existing pruning methods. While we focus on pruning as the main compression operator, we also give encouraging evidence that the CrAM update can be successfully adapted to other compression projections, such as quantization, and we plan to investigate this more closely in future work. We would like to explore whether prolonged CrAM-training would further enhance both the performance of the resulting dense model, as well as its robustness to one-shot compression. Finally, we are interested in leveraging in the CrAM update different methods developed for reducing the computational complexity of SAM, in order to improve the efficiency of our method.

## ACKNOWLEDGEMENTS

AP, EK, DA received funding from the European Research Council (ERC) under the European Union's Horizon 2020 research and innovation programme (grant agreement No 805223 ScaleML). AV acknowledges the support of the French Agence Nationale de la Recherche (ANR), under grant ANR-21-CE48-0016 (project COMCOPT). We further acknowledge the support from the Scientific Service Units (SSU) of ISTA through resources provided by Scientific Computing (SciComp)

## ETHICS AND REPRODUCIBILITY STATEMENTS

Our work is technical in nature, considering existing models and datasets, and thus does not pose any direct ethical concerns. With regards to reproducibility, we release an implementation of our optimizer and an example experimental harness.

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

APPENDIX

## A    THEORETICAL SUPPORT FOR THE CRAM UPDATE

In this section, we attempt to formally derive a generic training method whose purpose is to provide accurate models, which perform well even after being compressed. To understand why we can hope to achieve such guarantees, we first take a brief detour to the area of robust optimization.

### A.1    ROBUST OPTIMIZATION

Generally, practical training methods are based on versions of stochastic gradient descent attempting to minimize a loss function $L(\boldsymbol{\theta})$. However, $\boldsymbol{\theta}$ might turn out to be a bad solution as the landscape of $L$ in its neighborhood could contain large changes in value. To address this issue, one may attempt to flatten $L$ such that it is less sensitive to sharp drops in value localized around a very small region. To this extent, a standard robustification can be defined by

$$\widetilde{L}(\boldsymbol{\theta}) = \max_{\|\boldsymbol{\delta}\| \leq \rho} L(\boldsymbol{\theta} + \boldsymbol{\delta}), \tag{7}$$

which makes the value of $\widetilde{L}(\boldsymbol{\theta})$ take that of the largest value of $L$ given by perturbation of $\boldsymbol{\theta}$ within a ball of radius $\rho$. While this robustified function may seem well suited to generic training tasks, it is a priori unclear that it is amenable to optimization.

However, under certain conditions, we can efficiently optimize $\widetilde{L}$ by using a classical theorem in robust optimization due to Danskin (Danskin, 2012).

**Theorem 1.** *(Danskin) Let $\mathcal{C} \subseteq \mathbb{R}^m$ be a compact set, let a function $\phi : \mathbb{R}^n \times \mathcal{C} \to \mathbb{R}$ such that $\phi(\cdot, \boldsymbol{y})$ is continuously differentiable for every fixed $\boldsymbol{y} \in \mathcal{C}$ and $\nabla_{\boldsymbol{x}}\phi(\boldsymbol{x}, \boldsymbol{y})$ is continuous on $\mathbb{R}^n \times \mathcal{C}$, and let $\psi : \mathbb{R}^n \to \mathbb{R}$ be defined as*

$$\psi(x) = \max_{\boldsymbol{y} \in \mathcal{C}} \phi(\boldsymbol{x}, \boldsymbol{y}) \ .$$

*Then $\psi$ is locally Lipschitz continuous, directionally differentiable, and its directional derivatives satisfy*

$$d\psi(\boldsymbol{x}; \boldsymbol{d}) = \max_{\boldsymbol{y} \in \mathcal{C}^*} \boldsymbol{d}^\top \nabla_{\boldsymbol{x}}\phi(\boldsymbol{x}, \boldsymbol{y}) \ .$$

*where $\mathcal{C}^*(\boldsymbol{x})$ is the set of maximizers*

$$\mathcal{C}^*(\boldsymbol{x}) = \left\{ \boldsymbol{y}^* : \phi(\boldsymbol{x}, \boldsymbol{y}^*) = \max_{\boldsymbol{y} \in \mathcal{C}} \phi(\boldsymbol{x}, \boldsymbol{y}) \right\} \ .$$

*In particular, if for some $\boldsymbol{x} \in \mathbb{R}^n$ the set $\mathcal{C}^*(\boldsymbol{x}) = \{\boldsymbol{y}_{\boldsymbol{x}}^*\}$ is a singleton, then $\psi$ is differentiable at $\boldsymbol{x}$ and*

$$\nabla\psi(\boldsymbol{x}) = \nabla_{\boldsymbol{x}}\phi(\boldsymbol{x}, \boldsymbol{y}_{\boldsymbol{x}}^*) \ .$$

This shows that, under certain assumptions, we can obtain directional derivatives for $\widetilde{L}(\boldsymbol{\theta})$ by simply maximizing $L(\boldsymbol{\theta} + \boldsymbol{\delta})$ over $\boldsymbol{\delta} \in B_2(\rho)$.

**Corollary 2.** *Let $\widetilde{L}$ be defined as in Equation (7), and define*

$$\mathcal{C}^*(\boldsymbol{\theta}) = \left\{ \boldsymbol{\delta} : \|\boldsymbol{\delta}\| \leq \rho, L(\boldsymbol{\theta} + \boldsymbol{\delta}) = \max_{\|\boldsymbol{\delta}^*\| \leq \rho} L(\boldsymbol{\theta} + \boldsymbol{\delta}^*) \right\} \ ,$$

*and let $\overline{\boldsymbol{\delta}} \in \mathcal{C}^*(\boldsymbol{\theta})$. Provided that $L(\boldsymbol{\theta})$ is continuously differentiable, and $\boldsymbol{\theta}$ is not an articulation point for $\widetilde{L}$, $-\nabla L(\boldsymbol{\theta} + \overline{\boldsymbol{\delta}})$ is a descent direction for $\widetilde{L}(\boldsymbol{\theta})$ as long as it is nonzero.*

*Proof.* Let $\boldsymbol{h} = \nabla L(\boldsymbol{\theta} + \overline{\boldsymbol{\delta}})$. We apply Danskin's theorem for $\phi(\boldsymbol{\theta}, \boldsymbol{\delta}) = L(\boldsymbol{\theta} + \boldsymbol{\delta})$ and $\mathcal{C} = B_2(\rho)$. This shows that

$$d\widetilde{L}(\boldsymbol{\theta}; \boldsymbol{h}) = \sup_{\boldsymbol{\delta} \in \mathcal{C}^*(\boldsymbol{\theta})} \boldsymbol{h}^\top \nabla L(\boldsymbol{\theta} + \boldsymbol{\delta}) \geq \boldsymbol{h}^\top \nabla L(\boldsymbol{\theta} + \overline{\boldsymbol{\delta}}) = \boldsymbol{h}^\top \boldsymbol{h} \geq 0 \ .$$

Provided that $\boldsymbol{\theta}$ is not an articulation point for $\widetilde{L}$, we also have that $d\widetilde{L}(\boldsymbol{\theta}; -\boldsymbol{h}) = -d\widetilde{L}(\boldsymbol{\theta}; \boldsymbol{h}) \leq 0$, which concludes the proof. $\qquad\square$

### A.1.1 FROM ROBUST OPTIMIZATION TO SAM

Per Corollary 2, to obtain a descent direction it suffices to maximize $L(\boldsymbol{\theta} + \boldsymbol{\delta})$ over the set of perturbations satifying $\|\boldsymbol{\delta}\| \leq \rho$. In general, even when the underlying function $L$ is convex, this may be a difficult problem. Instead, one may simply attempt to obtain a good local maximizer of $L$ in a bounded region around $\boldsymbol{\theta}$. The simplest possible way to do so is by performing a step of *gradient ascent*, which can be regarded as a proxy for the maximization subproblem. Using this step, we immediately obtain the iteration:

$$\widetilde{\boldsymbol{\theta}}_t = \boldsymbol{\theta}_t + \frac{\rho}{\|\nabla L(\boldsymbol{\theta}_t)\|} \nabla L(\boldsymbol{\theta}_t), \quad \boldsymbol{\theta}_{t+1} = \boldsymbol{\theta}_t - \eta \nabla L(\widetilde{\boldsymbol{\theta}}_t), \tag{8}$$

which recovers the extrapolated SAM gradient step from Foret et al. (2021).

There is exhaustive research that has previously been done on robust optimization methods. For a comprehensive reference, we point the reader to Teo's PhD thesis (Teo, 2007).

### A.2 ROBUST OPTIMIZATION FOR COMPRESSIBLE MODELS

With the robust optimization framework in mind, we are ready to attempt implementing a similar scheme which exhibits robustness to compression.

To motivate the method, let us consider the post-training compression. After training the model to weights $\boldsymbol{\theta}_T$ we apply a one-shot compression method $C$ over some perturbation $\boldsymbol{\theta}_T + \boldsymbol{\delta}$ of the weights. This captures several iterative methods for compression, such as iterative pruning, where changes in weights are alternated with one-shot pruning methods.

If our goal is to make the loss after compression robust within a small neighborhood of perturbations $\boldsymbol{\delta}$, we can establish as a formal objective to minimize the robustified loss

$$L^{\text{CrAM}}(\boldsymbol{\theta}) := \max_{\boldsymbol{\delta}:\|\boldsymbol{\delta}\|\leq\rho} L(C(\boldsymbol{\theta} + \boldsymbol{\delta})), \tag{9}$$

for some magnitude $\rho$ of allowed perturbations. In our case we will focus on the case where these are bounded in $\ell_2$ norm, but this can be easily extended to other choices of the domain. Just as before, we can now attempt to minimize $L^{\text{CrAM}}$, or find a near-stationary point, by gradient descent. Using the robust optimization framework we may attempt to optimize it using Corollary 2 after replacing $L(\cdot)$ with $L(C(\cdot))$.

Naturally, this poses some obstacles in our case. The main one is the fact that it is not true that $L(C(\boldsymbol{\theta}))$ will generally be continuously differentiable, so the conditions required to obtain descent directions via an inner maximization loop are not satisfied. However, we can show that under certain conditions, continuous differentiability fails only at a set of points of measure 0.

**Definition 1.** *Let $S$ be a countable set, let $\{P_i\}_{i\in S}$ be a covering of $\mathbb{R}^n$ with convex sets, and let $S(\boldsymbol{x})$ denote the family of indices from $S$ for which $\boldsymbol{x} \in P_i$. Let a family of projection operators $\{\Pi_i\}_{i\in S}$, such that for any $\boldsymbol{x}$ the projections $\{\Pi_i(\boldsymbol{x})\}_{i\in S(\boldsymbol{x})}$ all map to the same point. We call a projective compression operator with respect to $\{\Pi_i\}_{i\in S}$ a mapping $C : \mathbb{R}^n \to \mathbb{R}^n$ such that*
$$C(\boldsymbol{x}) = \Pi_i(\boldsymbol{x}), \quad \text{for any } i \in S(\boldsymbol{x}).$$

For example, in the case of the *Top-K* compression operator, we can define a covering of $\mathbb{R}^n$ with sets $P$ such that all elements $x \in P$ share the indices $A \subseteq [n]$, $|A| = k$ for the Top-K elements in absolute value (with ties broken lexicographically), and furthermore, all elements from $P$ preserve the signs across $A$. It is clear that any $x \in \mathbb{R}^n$ belongs in some such set $P$, and since there are a finite number of subsets of size $k$ and of possible signs for the components in $[n]$, we have a finite covering. Assume, without loss of generality, that a set $P$ from the covering consists of elements for which the first $k$ components are the highest in absolute value, and the signs across these components are shared across all $x \in P$. Then, for any $\boldsymbol{x}, \boldsymbol{y} \in P$, $\lambda \in (0, 1)$ and $i \leq k$, we have that $|\lambda \boldsymbol{x}_i + (1 - \lambda)\boldsymbol{y}_i| = \lambda|\boldsymbol{x}_i| + (1 - \lambda)|\boldsymbol{y}_i|$ (since $\boldsymbol{x}_i$ and $\boldsymbol{y}_i$ share the same sign). Using that $\lambda|\boldsymbol{x}_i| + (1 - \lambda)|\boldsymbol{y}_i| \geq \lambda|\boldsymbol{x}_j| + (1 - \lambda)|\boldsymbol{y}_j|$, for any $k < j < n$, together with the triangle inequality, we obtain that $|\lambda \boldsymbol{x}_i + (1 - \lambda)\boldsymbol{y}_i| \geq |\lambda \boldsymbol{x}_j + (1 - \lambda)\boldsymbol{y}_j|$, for any $i \leq k$ and $j > k$. Therefore, any $P$ satisfying the conditions described above is a convex set. We can further define a projection for each subset $A$ of coordinates of cardinality $k$. Then, it is clear that for any given vector $\boldsymbol{x}$, the set $A \in S(\boldsymbol{x})$ iff the largest $k$ coordinates of $\boldsymbol{x}$ in absolute value (with ties broken lexicographically) are supported in $A$. Therefore, we can conclude that Top-K is a projective compression operator.

**Lemma 3** (Continuously differentiable functions induce few singularities after compression). *Let $L : \mathbb{R}^n \to \mathbb{R}$ be a continuously differentiable function, and let $C$ be a projective compression operator. Then the function $g(\boldsymbol{x}) := L(C(\boldsymbol{x}))$ is continuously differentiable everywhere except at a set of points of measure $0$. Furthermore, so is the robustified function $L^{\mathrm{CrAM}}(\boldsymbol{x}) := \max_{\|\boldsymbol{\delta}\| \leq \rho} L(C(\boldsymbol{x} + \boldsymbol{\delta}))$.*

*Proof.* First we note that the boundary of any convex set has measure zero by standard arguments in convex analysis (Lang, 1986). Since a countable union of sets of measure zero has measure zero, it follows that the union of the boundaries of $P_i$'s has measure zero. Now since $L$ is continuously differentiable, within any set $P_i$, we have that $g(\boldsymbol{x}) = L(\Pi_i(\boldsymbol{x}))$, and hence it remains continuously differentiable. Therefore the only region for which we can not argue about continuous differentiability is the complement of the union of interiors of $P_i$'s, $(\cup_i \mathrm{int} P_i)^c \subseteq \cup_i \partial P_i$ which is a set of measure zero. Since $g$ is well-behaved almost everywhere, all that remains to argue is that this is the same case with $L^{\mathrm{CrAM}}$.

For any fixed direction $\Delta\boldsymbol{\theta}$, we define the mapping

$$M(\boldsymbol{\theta}) = \Delta\boldsymbol{\theta}^\top \nabla g(\boldsymbol{\theta}) ,$$

and its robustification

$$\widetilde{M}(\boldsymbol{\theta}) = \max_{\|\boldsymbol{\delta}\| \leq \rho} M(\boldsymbol{\theta} + \boldsymbol{\delta}) = \max_{\|\boldsymbol{\delta}\| \leq \rho} \Delta\boldsymbol{\theta}^\top \nabla g(\boldsymbol{\theta} + \boldsymbol{\delta}) .$$

Hence the directional derivative w.r.t. $\Delta\boldsymbol{\theta}$ of $L^{\mathrm{CrAM}}(\boldsymbol{\theta})$ is discontinuous only when $\widetilde{M}(\boldsymbol{\theta})$ is discontinuous. Finally, we note that this almost never happens, as $M$ is continuous almost everywhere, and thus so must be $\widetilde{M}$. Thus, all directional derivatives are continuous except at a set of measure $0$, which concludes the proof. □

Finally, just like in the previous case, maximizing $L(C(\boldsymbol{\theta} + \boldsymbol{\delta}))$ over small perturbations is generally intractable. So we instead consider obtaining a good enough maximizer via a standard iterative method which has shown good performance in practice. More precisely we consider the projected gradient ascent method, which provides strong theoretical guarantees, even when the projection is performed onto non-convex domains (Peste et al., 2021). In the case where the compression operator represents magnitude pruning, this corresponds to the iterative hard thresholding (IHT) method, frequently employed in the sparse recovery literature.

To reach a good iterate within this specific domain we instead perform a single step of (projected) gradient ascent, which matches the IHT iteration:

$$\widetilde{\boldsymbol{\theta}}_t = C\left(\boldsymbol{\theta}_t + \rho \cdot \nabla L(\boldsymbol{\theta}_t)\right) . \tag{10}$$

We have therefore obtained a re-derivation of the CrAM update in Equation 2. We note that in our experiments we use a fixed step size $\rho$ for the interpolation step, instead of normalizing the gradients. Similar to previous work (Andriushchenko & Flammarion, 2022), we observed that normalizing the gradients did not result in significant difference in the results, and for simplicity we have omitted the normalization step in our experiments.

Furthermore, we note that the analysis above holds also for the CrAM$^+$. Namely, we can rewrite the CrAM$^+$ loss as $L^{\mathrm{CrAM}^+} = \max_{\|\boldsymbol{\delta}\| \leq \rho}(L(C(\boldsymbol{\theta} + \boldsymbol{\delta})) + L(\boldsymbol{\theta}))$, and use Lemma 3 to obtain that $L^{\mathrm{CrAM}^+}$ is continuously differentiable almost everywhere, and so are its directional derivatives.

## B IMAGE CLASSIFICATION HYPERPARAMETERS

**Hyperparameters for CIFAR-10 experiments** We train ResNet20 models for 200 epochs, using SGD with momentum and weight decay, and a cosine learning rate scheduler, with a learning rate warm-up of 5 epochs. Additionally, we trained the baseline model for twice as many epochs, to match the number of backpropagation steps of SAM and CrAM. To determine the value of the hyperparameter $\rho$, we performed a grid search over values in the range $0.01 - 0.2$, using a $90\% - 10\%$ train-validation split and found 0.1 and 0.15 to be the best values for SAM and CrAM$^+$-Multi, respectively (achieving highest validation accuracy). After finding the best value of $\rho$, we retrained using the entire training set, and starting from 3 different random seeds, and report the final accuracy after 200 epochs of training. We follow a very similar training recipe and hyperparameter search for ResNet18 and VGG experiments, but train instead of 180 epochs.

**Hyperparameters for ImageNet experiments** For our ImageNet experiments, we use standard data augmentation, and we train the models using SGD for 100 epochs, with batch size 512, momentum 0.9, and weight decay 0.0001. The learning rate is linearly increased for the first 5 epochs until it reaches a maximum value of 0.2, after which it is decreased at each epoch, using a cosine scheduler. To determine the value of the hyperparameter $\rho$, we search over a small grid, by training 90% of ImageNet using CrAM-k50, and using the remaining 10% of the dataset for validation. We have found $\rho = 0.05$ to give good results for CrAM-k50, and we have kept this value for all our other CrAM experiments. For SAM, we also use $\rho = 0.05$, which is the standard value recommended by the authors of Foret et al. (2021).

## C   ADDITIONAL IMAGE CLASSIFICATION EXPERIMENTS

### C.1   ABLATION STUDY FOR ALTERNATIVE UPDATES

**Comparison between CrAM and C-SAM.** We investigate the importance of individual components from the CrAM update by comparing against other similar updates, on the CIFAR-10 dataset, using a ResNet20 model. One such update can be obtained by following closely the derivations for SAM (Foret et al., 2021). We assume $\|\boldsymbol{\delta}\| \leq \rho$, define $h(\boldsymbol{x}) := L(C(\boldsymbol{x}))$ (with $C$ the Top-K operator), and the loss $\max_{\|\boldsymbol{\delta}\| \leq \rho} h(\boldsymbol{\theta} + \boldsymbol{\delta})$. We can assume that for a small enough $\rho$ the function $h(\boldsymbol{\theta} + h)$ is differentiable in $\boldsymbol{\theta}$ (for example, the perturbation is small enough such that the Top-K mask stays constant). By using a first-order Taylor approximation of $h(\boldsymbol{\theta} + \boldsymbol{\delta})$ around $\boldsymbol{\theta}$, together with the quadratic constraint for $\boldsymbol{\delta}$, we obtain $\boldsymbol{\delta} = \rho \cdot \frac{\nabla L(C(\boldsymbol{\theta}))}{\|\nabla L(C(\boldsymbol{\theta}))\|}$. This enables us to define the compressed-SAM (C-SAM) update as:

$$\text{C-SAM:} \qquad \boldsymbol{\theta}_{t+1} = \boldsymbol{\theta}_t - \eta \nabla L \left( C \left( \boldsymbol{\theta}_t + \rho \frac{\nabla L(C(\boldsymbol{\theta}_t))}{\|\nabla L(C(\boldsymbol{\theta}_t))\|} \right) \right) . \tag{11}$$

We train C-SAM using sparsified gradients, for both the intermediate interpolation step, as well as in the final weight update. Namely, we always use the approximation: $\nabla L(C(\phi)) \approx M_\phi \cdot \nabla L(M_\phi \cdot \phi)$ for parameter $\phi$, where $M_\phi$ is the Top-K mask of $\phi$. To determine the value of the interpolation step $\rho$ in C-SAM and CrAM, we perform a grid search over a 90-10% train/validation split of CIFAR-10; we re-run from 3 different seeds, using the best configurations and the entire training set, and report the test accuracy after 200 epochs of training. For fairness, we compare C-SAM with CrAM, and not CrAM$^+$, and for both we use sparsified gradients. For C-SAM-Multi and CrAM-Multi, we choose the sparsity levels during training uniformly at random among values in the interval $30\% - 90\%$.

The results in Table 7 show that C-SAM can be more robust at higher sparsity levels, but with the cost of an accuracy drop for the dense models. Moreover, the dense model can be improved using CrAM$^+$ with no additional cost, whereas for C-SAM such a modification would require a computational overhead. We additionally observed that the dense C-SAM-k70 model can be improved to almost match the accuracy of CrAM-k70, by not sparsifying the gradients in the interpolation; however, this also decreased the accuracy after one-shot pruning at higher sparsity, to a level similar to CrAM.

| Method | Dense | 50% Sparse | 60% Sparse | 70% Sparse | 80% Sparse | 90% Sparse |
|---|---|---|---|---|---|---|
| CrAM-k50 | $92.8 \pm 0.1$ | $92.8 \pm 0.1$ | $92.7 \pm 0.0$ | $92.0 \pm 0.1$ | $89.7 \pm 0.2$ | $73.5 \pm 2.4$ |
| C-SAM-k50 | $92.6 \pm 0.3$ | $92.6 \pm 0.3$ | $92.5 \pm 0.3$ | $92.0 \pm 0.2$ | $90.6 \pm 0.2$ | $81.0 \pm 1.0$ |
| CrAM-k70 | $92.4 \pm 0.2$ | $92.4 \pm 0.2$ | $92.4 \pm 0.2$ | $92.3 \pm 0.2$ | $91.6 \pm 0.1$ | $81.1 \pm 1.3$ |
| C-SAM-k70 | $91.4 \pm 0.0$ | $91.4 \pm 0.0$ | $91.4 \pm 0.0$ | $91.3 \pm 0.1$ | $91.0 \pm 0.1$ | $85.3 \pm 0.5$ |
| CrAM-Multi | $92.6 \pm 0.2$ | $92.5 \pm 0.2$ | $92.4 \pm 0.4$ | $92.3 \pm 0.2$ | $91.9 \pm 0.2$ | $90.5 \pm 0.2$ |
| C-SAM-Multi | $92.5 \pm 0.2$ | $92.5 \pm 0.1$ | $92.6 \pm 0.2$ | $92.5 \pm 0.2$ | $92.1 \pm 0.2$ | $91.0 \pm 0.2$ |

**Table 7:** (CIFAR-10/ResNet20) Comparison between CrAM and C-SAM. Test accuracy for the dense models and sparse models after one-shot pruning. We report the best value between the accuracy before and after BNT with 1000 training samples.

**Importance of the extra-gradient step.** Furthermore, we explore the importance of the extra-gradient step in the CrAM update. Notably, we investigate whether not using the extra-gradient achieves a similar effect to CrAM training. The removal of the extra gradient step would correspond to the following equation:

$$\text{Top-K:} \qquad \boldsymbol{\theta}_{t+1} = \boldsymbol{\theta}_t - \eta \nabla L(C(\boldsymbol{\theta}_t)) . \tag{12}$$

Since the compression we use in our experiments is the Top-K sparsification, we simply call this update "Top-K". This update has been previously studied in Lin et al. (2020), where the dense gradients, computed with respect to the sparse parameters, are used in the model updated. Generally, we have training instability using this update, particularly at high sparsity. However, incorporating the optimization of the dense model, as well as sparsified gradients for the compressed parameters, greatly improved the stability and overall quality of the resulting models. These changes resulted in an update close to CrAM$^+$ and of the same computational complexity, which will be referred to as Top-K$^+$:

$$\text{Top-K}^+ : \qquad \boldsymbol{\theta}_{t+1} = \boldsymbol{\theta}_t - \eta(\nabla L(\boldsymbol{\theta}_t) + M_{\widetilde{\boldsymbol{\theta}}_t} \cdot \nabla L(\widetilde{\boldsymbol{\theta}}_t)), \qquad (13)$$

where $\widetilde{\boldsymbol{\theta}}_t = C(\boldsymbol{\theta}_t)$ and $M_{\widetilde{\boldsymbol{\theta}}_t}$ is its mask after applying Top-K.

| Method | Dense | 50% Sparse | 60% Sparse | 70% Sparse | 80% Sparse | 90% Sparse |
|---|---|---|---|---|---|---|
| CrAM$^+$-k50 | $93.1 \pm 0.1$ | $93.1 \pm 0.1$ | $93.0 \pm 0.1$ | $92.3 \pm 0.2$ | $89.2 \pm 0.2$ | $71.1 \pm 1.5$ |
| Top-K$^+$-k50 | $92.7 \pm 0.1$ | $92.6 \pm 0.0$ | $92.6 \pm 0.1$ | $91.6 \pm 0.1$ | $86.5 \pm 0.3$ | $56.7 \pm 1.0$ |
| CrAM$^+$-k70 | $92.8 \pm 0.3$ | $92.7 \pm 0.2$ | $92.7 \pm 0.1$ | $92.7 \pm 0.0$ | $91.9 \pm 0.$ | $80.8 \pm 1.4$ |
| Top-K$^+$-k70 | $92.7 \pm 0.1$ | $92.4 \pm 0.2$ | $92.3 \pm 0.2$ | $92.4 \pm 0.1$ | $91.0 \pm 0.3$ | $72.6 \pm 2.8$ |
| CrAM$^+$-Multi | $93.2 \pm 0.1$ | $93.2 \pm 0.1$ | $93.0 \pm 0.1$ | $92.8 \pm 0.2$ | $92.4 \pm 0.1$ | $90.1 \pm 0.2$ |
| Top-K$^+$-Multi | $92.5 \pm 0.1$ | $92.4 \pm 0.1$ | $92.3 \pm 0.1$ | $92.2 \pm 0.2$ | $91.7 \pm 0.2$ | $90.0 \pm 0.2$ |

**Table 8:** (CIFAR-10/ResNet20) Comparison between CrAM$^+$ and Top-K$^+$. Test accuracy for the dense models and sparse models after one-shot pruning. For all sparse results we report the best value between the accuracy before and after BNT with 1000 training samples.

The results of the comparison between CrAM$^+$ and Top-K$^+$ are presented in Table 8 and show that CrAM models tend to have higher accuracy for the dense models, as well as to be more robust to one-shot pruning at high sparsity (e.g. CrAM$^+$ and Top-K$^+$ trained with 50% or 70% sparsity, and one-shot pruned to 80% and 90% sparsity).

The comparison between CrAM and C-SAM or Top-K shows that, although all methods can achieve good results with one-shot pruning, CrAM-trained models have the best trade-off between preserving (or improving) the dense model accuracy, while having good performance after one-shot pruning, at a range of sparsity levels.

**CrAM vs. SAM on compressed models.** We briefly highlight the differences between the CrAM update and simply applying SAM to compressed models. First, we note that the two objectives are different: while CrAM optimizes against the largest perturbation of the *dense model*, to minimize its effect after compression, training SAM on sparse models would be equivalent to optimizing against the largest perturbation on the *sparse model*. Namely, the SAM loss on compressed models could be written as $\max_{\|\boldsymbol{\delta}\| \leq \rho} L(C(\boldsymbol{\theta}) + \boldsymbol{\delta})$, with the intermediate update $\widetilde{\boldsymbol{\theta}} = C(\boldsymbol{\theta}) + \rho \cdot \frac{\nabla L(C(\boldsymbol{\theta}))}{\|\nabla L(C(\boldsymbol{\theta}))\|}$. In practice, applying SAM to compressed models could only ensure generalization for the compressed models, whereas with CrAM (and, in particular, CrAM$^+$) we can explicitly optimize the dense model, and ensure that it is robust to one-shot pruning.

## C.2 Importance of Sparse Gradients

For all our image classification experiments, we observed an improvement in the robustness to post-training one-shot pruning, when using a different straight-through estimator for the gradient $\nabla_\theta L(\widetilde{\boldsymbol{\theta}}_t)$, where $\widetilde{\boldsymbol{\theta}}_t = C(\boldsymbol{\theta}_t + \rho \cdot \nabla L(\boldsymbol{\theta}_t))$. Namely, instead of by-passing the Top-K operator in the gradient, and estimating $\nabla_{\boldsymbol{\theta}} L(\widetilde{\boldsymbol{\theta}}_t) \approx \nabla_{\boldsymbol{\theta}} L(\boldsymbol{\theta})|_{\boldsymbol{\theta}=\widetilde{\boldsymbol{\theta}}_t}$, we can assume that the masks $M_t$ of $\widetilde{\boldsymbol{\theta}}_t$ change very little during training. This would allow us to use the approximation $\nabla_{\boldsymbol{\theta}} L(\widetilde{\boldsymbol{\theta}}_t) \approx M_t \cdot \nabla_{\boldsymbol{\theta}} L(\boldsymbol{\theta})|_{\boldsymbol{\theta}=\widetilde{\boldsymbol{\theta}}_t}$. Please see Section 3.2 and Appendix Section A for more details. Training both CrAM and CrAM$^+$ with this new approximation for the CrAM loss gradient (which will be referred to as "sparse gradient") led to an improvement in the robustness to one-shot pruning, particularly at higher sparsity levels. Furthermore, our intuition that the masks are fairly constant during training is also confirmed experimentally: on CIFAR-10/ResNet20, trained with CrAM$^+$-k70, the difference between consecutive $\boldsymbol{\theta}_t$ masks was lower than 0.6%. Interestingly, using sparse gradient under the same setup, encouraged more diversity in the masks, with the difference between them at later training stages increasing to around 2%. We speculate this could be a potential reason for the improved robustness

to pruning. Another aspect observed on CIFAR-10 experiments is that using sparse gradients tends to decrease the dense model accuracy, when training CrAM at lower sparsity; for example, the dense model for CrAM-k50 reached 93.4% accuracy, which decreased to 92.8% when using sparse gradients. For this reason, on ImageNet we only experimented with the dense version of CrAM-k50. Nonetheless, using sparse gradients improved the robustness to pruning in all cases. For a better illustration of all these effects, we provide the results obtained with these different versions of CrAM on ImageNet, in Table 9 for one-shot unstructured global magnitude pruning and in Table 10 for semi-structured N:M pruning.

| Model | Dense | Sparsity | | | |
|---|---|---|---|---|---|
| | | 50% | 70% | 80% | 90% |
| CrAM-k70 | 75.7 | 76.3 | 76.3 | 73.4 | 53.2 |
| CrAM$^+$-k70 | **77.3** | **77.3** | 76.8 | 73.9 | 51.9 |
| CrAM$^+$-k70 (SG) | 77.2 | 77.2 | **77.2** | **76.3** | **62.1** |
| CrAM-Multi | 75.2 | 75.2 | 75.2 | 74.5 | 73.3 |
| CrAM$^+$-Multi | 76.4 | 76.4 | 76.1 | 74.9 | 73.1 |
| CrAM$^+$-Multi (SG) | **77.3** | **77.2** | **77.0** | **75.8** | **74.7** |

**Table 9:** (ImageNet/ResNet50) Dense and one-shot pruning (+BNT) results. CrAM$^+$ with sparse gradients (SG) improves the accuracy of the dense model, and its robustness to one-shot pruning.

| Model | Dense | Sparsity Pattern | |
|---|---|---|---|
| | | 2:4 | 4:8 |
| CrAM-N:M | 75.2 | 76.0 | 76.2 |
| CrAM$^+$-N:M | 77.1 | 76.2 | 76.7 |
| CrAM$^+$-N:M (SG) | 77.3 | 77.0 | 77.2 |
| SR-STE | - | 77.0 | 77.4 |

**Table 10:** (ImageNet/ResNet50) Dense and semi-structured one-shot pruning (+BNT) results. CrAM$^+$-N:M with sparse gradients (SG) improves the accuracy of the dense model, and its robustness to N:M pruning.

### C.3 Variability of Batch Norm Tuning Results

We emphasize that CrAM relies on a small calibration set of training samples to correct the Batch Norm statistics, namely running mean and variance, after pruning, particularly at high sparsity. We call this procedure Batch Norm Tuning (BNT). To ensure that the accuracies we report for sparse models are stable under the choice of the calibration set, we perform 10 independent trials of BNT, on 10 randomly chosen subsets of 1000 training samples, for each model and for different sparsity levels. The results of this experiment are presented in Table 11, which also contains the raw numbers used in Figure 1. Notice that the accuracy after one-shot pruning and BNT is very stable, with respect to the choice of the calibration set. In particular, for the CrAM$^+$ model, the standard deviation is $\leq 0.1\%$ across all sparsity levels considered. We also report the raw one-shot pruning accuracy, before BNT, for CrAM models in Table 12.

| Model | Dense | Sparsity | | | | |
|---|---|---|---|---|---|---|
| | | 50% | 60% | 70% | 80% | 90% |
| Baseline | 77.22 | $75.87 \pm 0.09$ | $73.82 \pm 0.07$ | $68.86 \pm 0.08$ | $51.96 \pm 0.27$ | $8.57 \pm 0.11$ |
| SAM | 77.35 | $76.47 \pm 0.04$ | $75.11 \pm 0.1$ | $71.87 \pm 0.07$ | $60.20 \pm 0.13$ | $18.25 \pm 0.18$ |
| CrAM-k50 | 77.48 | $77.3 \pm 0.07$ | $76.61 \pm 0.05$ | $74.77 \pm 0.08$ | $68.23 \pm 0.11$ | $33.04 \pm 0.16$ |
| CrAM$^+$-k70 | 77.32 | $77.22 \pm 0.05$ | $77.1 \pm 0.05$ | $77.15 \pm 0.05$ | $76.3 \pm 0.08$ | $61.92 \pm 0.11$ |
| CrAM$^+$-Multi | 77.28 | $77.24 \pm 0.06$ | $77.05 \pm 0.05$ | $77.0 \pm 0.04$ | $75.8 \pm 0.07$ | $74.74 \pm 0.04$ |

**Table 11:** (ImageNet/ResNet50) Validation accuracy for the dense models, and after one-shot pruning using global magnitude pruning, followed by BNT on 1000 samples. The results for one-shot pruning are the mean accuracies, and their standard deviations, when BNT is performed on 10 different random calibration sets, of 1000 training samples each.

| Model | Dense | Sparsity | | | | |
|---|---|---|---|---|---|---|
| | | 50% | 60% | 70% | 80% | 90% |
| Baseline | 77.22 | 74.35 | 68.9 | 46.36 | 2.0 | 0.1 |
| SAM | 77.35 | 75.02 | 70.4 | 52.8 | 3.66 | 0.11 |
| CrAM-k50 | 77.48 | 75.91 | 73.54 | 63.05 | 13.59 | 0.16 |
| CrAM$^+$-k70 | 77.32 | 77.03 | 76.6 | 76.3 | 72.6 | 3.6 |
| CrAM$^+$-Multi | 77.28 | 76.23 | 74.87 | 75.69 | 72.32 | 52.25 |

**Table 12:** (ImageNet/ResNet50) Validation accuracy for the dense models, and after one-shot pruning using global magnitude, before BNT.

## C.4    RESULTS WITH UNIFORM SPARSITY

In this section we show that CrAM models can be trained to be robust to different sparsity distributions, such as uniform. We train CrAM$^+$-Multi models for ImageNet/ResNet50 under the same setup as in Section 4.1, but applying instead the Top-K operator at uniform sparsity across all prunable parameters (i.e. excluding BatchNorm and biases), while keeping the first and last layers dense. The resulting dense model achieves 77.1% accuracy, while one-shot uniform pruning at 80% and 90% sparsities gives, after BNT, 75.6% and 75.1% accuracy, respectively. Moreover, this model is also robust to one-shot pruning using global magnitude (e.g. 75.4% accuracy at 80% sparsity). Conversely, CrAM$^+$-Multi trained with global magnitude is robust to one-shot pruning using uniform magnitude (e.g. 76.9% and 75.9% accuracy at 70% and 80% sparsity, respectively). This suggests that CrAM-trained models can be robust to one-shot pruning using sparsity distributions different from the ones used during training.

## C.5    CRAM FOR N:M SPARSITY PATTERNS

In this section, we show our full results regarding the robustness of CrAM models against semi-structured N:M sparsity patterns, where out of each block of M weights, N are sparse. In particular, the 2:4 pattern is supported on modern Ampere NVIDIA architectures, where it has been shown to provide speed-ups (Mishra et al., 2021). We train CrAM$^+$ models with the N:M pattern, by randomly choosing at each optimization step between the 2:4 or 4:8 projections; this model will be referred to as "CrAM$^+$-N:M". Similar to the previous experiments, we also use sparse gradients for the pruned model perturbation and have found this to have a positive impact on the one-shot pruned models. In Table 13 we show the one-shot pruning results (after BNT with 1000 samples). Note that CrAM$^+$-N:M models do not lose accuracy when they are pruned one-shot using the 2:4 and 4:8 patterns, which is competitve with state-of-the-art methods for training N:M sparse models, such as SR-STE (Zhou et al., 2021); however, SR-STE requires a different training run for each sparsity profile. CrAM$^+$-N:M model is also robust to one-shot pruning using unstructured patterns, at moderate sparsity levels; for example, the results in Table 14 show that models trained with CrAM$^+$-N:M can be pruned one-shot to 70% sparsity with a minor accuracy loss, compared to the dense baseline.

| Model | Dense | 2:4 | 4:8 |
|---|---|---|---|
| Dense | 77.3 | 66.1 | 69.1 |
| SAM | 77.4 | 69.4 | 71.8 |
| CrAM-k50 | 77.5 | 72.1 | 73.3 |
| CrAM$^+$-N:M | 77.3 | 77.0 | 77.2 |
| SR-STE | - | 77.0 | 77.4 |

**Table 13:** Accuracy (%) after one-shot pruning (+BNT) using semi-structured 2:4 and 4:8 patterns.

| Top-K | 50% | 70% |
|---|---|---|
| Global | 77.3 | 76.5 |
| Uniform | 77.3 | 76.5 |

**Table 14:** [CrAM$^+$-N:M] Accuracy (%) after one-shot pruning (+BNT), using unstructured sparsity

## C.6    CRAM WITH INFREQUENT MASK UPDATES

As previously mentioned, when training image models with CrAM$^+$, using sparse gradients of the intermediate model perturbation substantially improved both the accuracy of the dense final model, as well as after one-shot pruning. One hypothesis that would enable this approximation of the gradient would be that the masks of the compression perturbation change very little during training. We test the validity of this hypothesis by training a version of CrAM$^+$ that only does infrequent mask updates. Namely, the masks obtained from the Top-K compression operator are kept fixed for a number of consecutive $\tau$ training iterations. In the case of CrAM-Multi, the masks for each sparsity level are changed each after $\tau$ iterations with the corresponding sparsity target. We note that infrequent mask updates improve the practical training time of CrAM, as the Top-K operator can be skipped for most iterations. This brings the cost of an average CrAM$^+$ iteration to the same level as a SAM iteration, though we note that in theory, with specialized hardware, the computational cost of CrAM can be further decreased due to the presence of sparse operators. We experiment using the same setup for training ImageNet on ResNet50 with CrAM$^+$-Multi, with global magnitude pruning at sparsity $k \in \{50\%, 70\%, 90\%\}$, and perform two separate runs, by varying the mask update frequency to $\tau \in \{20, 100\}$ iterations. As before, we also use BNT after pruning, on a subset of 1000 training samples, consisting of one example per class. The results presented in Table 15 show that using

infrequent mask updates only has a small impact on the initial results; namely, we observe that the results after one-shot pruning are slightly worse compared to the default ones, particularly at higher sparsity. In particular, the results for 80% sparsity have decreased the most, a level which was not explicitly used during training. We also repeated the same experiment in the CrAM-finetuning setup for ResNet50, with $\tau = 100$ iterations, and observed that results after one-shot pruning in fact improved slightly. This is particularly encouraging, as using CrAM only for finetuning is a more attractive use-case, due to the reduced computational costs.

| Frequency | Sparsity (%) | | | | |
|---|---|---|---|---|---|
| $\tau$ | 0 | 50 | 70 | 80 | 90 |
| 1 | 77.3 | 77.2 | 77.1 | 75.9 | 74.8 |
| 20 | 77.4 | 77.4 | 77.2 | 75.5 | 74.8 |
| 100 | 77.3 | 77.3 | 76.9 | 75.3 | 74.5 |

**Table 15:** (ImageNet/ResNet50) Validation accuracy (%) for the dense and sparse CrAM$^+$-Multi models trained with sparse perturbed gradients, using infrequent mask updates of the global Top-K operator.

## C.7 RESULTS ON QUANTIZATION

We have shown through previous experiments that CrAM can be successfully used with the Top-K operator to obtain models that preserve or improve the dense baseline's accuracy, while also being robust to post-training one-shot pruning, to multiple sparsity levels. In this section we show encouraging evidence that CrAM can also be adapted to work with quantization. Specifically, we use CrAM$^+$ where the compression operator $C$ is the symmetric per-channel weight quantization to 4 bits (round to nearest integer), and finetune pretrained Torchivison ResNet18 and ResNet50 ImageNet models, for 10 epochs, using the same hyperparameters as in the previous experiments for sparsity.

| Model | ResNet18 | | ResNet50 | |
|---|---|---|---|---|
| | Dense | 4 Bits | Dense | 4 Bits |
| Baseline | 69.8 | 66.2 | 76.1 | 74.1 |
| SAM | **70.5** | 67.1 | **76.9** | 74.8 |
| CrAM$^+$-4Bits | 70.1 | **69.3** | 76.7 | **75.8** |

**Table 16:** [ImageNet] Validation acc. (%) for the dense models, and after symmetric per-channel 4 bits quantization (+BNT).

## C.8 COMPARISON WITH OTHER METHODS ON CIFAR-10

In this section we provide additional results accompanying those presented in Section 4.3. Namely, we provide comparison between one-shot pruning CrAM$^+$-Multi vs. standard dense baselines (SGD, SAM), and we provide numbers before and after BNT for sparse models on VGG-16 and ResNet18.

In Table 17 we show the accuracy of the dense baseline, SAM and CrAM$^+$-Multi, trained using sparsity levels sampled uniformly at random in the range $30\% - 90\%$ on ResNet20, before and after one-shot pruning at different sparsities. The results after one-shot pruning are presented after BNT over a random subset of 1000 train samples, over 100 batches. We note there are small variations in the results after BNT, due to the choice of the random calibration set. These variations are small ($\pm 0.1/0.2\%$) for CrAM$^+$-Multi models, across all sparsity levels considered, but they are larger for the one-shot pruned dense baselines at high sparsity (e.g. 80% and 90%). Moreover, the accuracy before BNT is still high for CrAM at lower sparsity levels (e.g. 91.9% at 70% sparsity), but it degrades at high sparsity (e.g. 50.5% at 90% sparsity). We believe that this is only due to the BatchNorm statistics, which are adapted to the dense model during training, but they no longer reflect the distribution shift after weight pruning. This is confirmed by the fact that 90% sparse models improve to over 90% test accuracy after only a few iterations of BNT, and are very robust to the choice of the calibration set.

Moreover, we show the extended results of CrAM$^+$-k95 discussed in Section 4.3, before and after BNT, on ResNet18 and VGG-16. From Table 18 we can see that one-shot pruning CrAM$^+$-k95 without BNT preserves accuracy up to 80% sparsity, after which BNT is required to correct the BatchNorm statistics. Remarkably, the VGG-16 models at 97% and 98% sparsity have very low

| Model | Dense | Sparsity | | | | |
| --- | --- | --- | --- | --- | --- | --- |
| | | 50% | 60% | 70% | 80% | 90% |
| Baseline | $93.0 \pm 0.1$ | $92.2 \pm 0.0$ | $91.0 \pm 0.3$ | $88.0 \pm 0.2$ | $78.0 \pm 1.1$ | $45.8 \pm 3.0$ |
| SAM | $\mathbf{93.5 \pm 0.1}$ | $92.8 \pm 0.2$ | $92.4 \pm 0.0$ | $90.7 \pm 0.3$ | $85.2 \pm 0.4$ | $54.6 \pm 1.7$ |
| CrAM$^+$-Multi | $93.2 \pm 0.1$ | $93.2 \pm 0.1$ | $93.1 \pm 0.1$ | $92.9 \pm 0.1$ | $92.4 \pm 0.1$ | $90.3 \pm 0.1$ |

**Table 17:** (CIFAR-10/ResNet20) Test acc. (%) for the dense models, and after one-shot pruning (+BNT). The baseline is the model after SGD training. For all models we apply one-shot pruning at different sparsity (+BNT), but no additional retraining. Results are averaged across 3 runs from different seeds.

accuracy, which is improved greatly by BNT. Furthermore, also in this highly sparse regimes the accuracy is very robust with respect to the choice of the calibration set for BNT.

| Architecture | BNT | Sparsity | | | | | | |
| --- | --- | --- | --- | --- | --- | --- | --- | --- |
| | | 50% | 80% | 90% | 93% | 95% | 97% | 98% |
| ResNet18 | No | $95.7\pm0.1$ | $95.3\pm0.2$ | $93.6\pm0.7$ | $92.0\pm1.4$ | $89.8\pm2.2$ | $81.4\pm6.3$ | $48.7\pm5.8$ |
| | Yes | $95.6\pm0.0$ | $95.7\pm0.0$ | $95.5\pm0.1$ | $95.5\pm0.1$ | $95.5\pm0.1$ | $95.2\pm0.0$ | $94.5\pm0.3$ |
| VGG-16 | No | $94.2\pm0.1$ | $93.9\pm0.2$ | $86.7\pm1.6$ | $48.5\pm1.7$ | $19.8\pm15.4$ | $16.0\pm10.2$ | $12.4\pm4.1$ |
| | Yes | $94.2\pm0.1$ | $94.2\pm0.1$ | $94.0\pm0.1$ | $94.0\pm0.2$ | $94.1\pm0.1$ | $93.8\pm0.2$ | $93.0\pm0.2$ |

**Table 18:** (CIFAR-10) Test acc. (%) for the sparse models obtained with one-shot-pruning from CrAM$^+$-k95, before and after BNT. Results are averaged across 3 runs from different seeds.

# D  LANGUAGE MODELS - REPRODUCIBILITY AND HYPERPARAMETERS

To ease reproducibility of our results, we conduct all of our experiments with the popular open-source libraries: Transformers (Wolf et al., 2020), and SparseML (Kurtz et al., 2020). We use the publicly available datasets via Lhoest et al. (2021), and focus on the BERT-base (Devlin et al., 2019) as it is one of the most commonly used language models. It is composed of 12 identical transformer layers with 110M parameters. Following community standards, we prune all weights of the encoder part (85M) and report sparsities relative to this number.

**General setup.** Our SQuADv1.1 fine-tuning recipe with Adam, SAM and CrAM mostly follows the already established hyper-parameters (Devlin et al., 2019; Wolf et al., 2020): start from the pretrained `bert-base-uncased` (available for download at https://huggingface.co/bert-base-uncased), `batch-size=16`, `max-sequence-length=384`, `doc-stride=128`.

**Adam, SAM, and CrAM optimization.** For other hyper-parameters we conduct a grid search for each optimizer independently over the following values: `learning-rate` $\in \{3e{-}5, 5e{-}5, 8e{-}5\}$; `num-train-epochs` $\in \{2, 3\}$ for SAM and CrAM, and `num-train-epochs` $\in \{2, 3, 4, 6\}$ for Adam (we allow 2x more epochs for fairness to SAM and CrAM); `label-smoothing-factor` $\in \{0.0, 0.1, 0.2\}$. We freeze the embedding layer in all experiments. To determine the value of the hyperparameter $\rho$, we performed a grid search over values in the range $1e{-}4$ to $1e{-}1$. For each optimizer we pick the set of hyperparameters that produces the best results after one-shot magnitude pruning to 50% sparsity, and they are as follows:

- Adam: `num-train-epochs=2`, `learning-rate=8e−5`, `label-smoothing-ratio=0.1`

- SAM: `num-train-epochs=2`, `learning-rate=8e−5`, `label-smoothing-ratio=0.0`, $\rho=0.01$

- CrAM (all CrAM runs use the same set of hyperparameters): `num-train-epochs=3`, `learning-rate=8e−5`, `label-smoothing-ratio=0.2`, $\rho=0.005$

At each CrAM optimization step we apply Top-K (i.e. magnitude) sparsification over all layers uniformly.

**One-shot pruning.** We apply one-shot pruning with two different pruners: magnitude and oBERT. For one-shot magnitude pruning we impose uniform sparsity distribution over all layers. For one-shot oBERT pruning we adopt the suggested set of hyper-parameters by authors, which we briefly describe here for completeness: 1024 gradients, dampening $1e{-}7$, block-size 50, 4 recomputations, global

sparsity distribution over all layers. For more details please refer to the oBERT paper (Kurtic et al., 2022).

**Sparse fine-tuning of one-shot pruned models.** We fine-tune one-shot oBERT-pruned models with the fixed sparsity mask and Adam optimizer. To identify the best set of hyperparameters for fine-tuning of the sparse model, we conduct a grid search over the following parameters: `learning-rate` $\in \{3e{-}5, 5e{-}5, 8e{-}5, 1e{-}4\}$, `num-train-epochs` $\in \{1, 2\}$, `label-smoothing-ratio` $\in \{0.0, 0.2\}$, `warmup-ratio` $\in \{0.0, 0.1\}$. We freeze the embedding layer and employ early-stopping technique to prevent overfitting.

| Model | Dense | Sparsity | | | |
|---|---|---|---|---|---|
| | | 50% | 60% | 70% | 80% |
| p(Adam) = 0.0 | 88.7 | **88.3** | **88.1** | **86.8** | 82.5 |
| p(Adam) = 0.1 | 87.6 | 87.4 | 87.4 | 86.5 | **84.0** |
| p(Adam) = 0.3 | 87.5 | 87.5 | 87.2 | 86.5 | 83.6 |
| p(Adam) = 0.5 | 87.8 | 87.7 | 87.2 | 86.4 | 83.0 |
| p(Adam) = 0.8 | 87.0 | 87.1 | 86.8 | 85.2 | 79.1 |

**Table 19:** (SQuADv1.1/BERT-base) Validation F1 score of models optimized with CrAM$^+$-Multi where at each step with probability $p(\mathrm{Adam})$ the standard Adam step is applied instead of the CrAM$^+$-Multi step.

| Model | 4-cores, batch-size=1 | | 16-cores, batch-size=128 | |
|---|---|---|---|---|
| | Throughput (items/sec) | Speed-up | Throughput (items/sec) | Speed-up |
| Dense | 4.0 | 1.0x | 14.2 | 1.0x |
| 50% sparse | 4.5 | 1.1x | 18.0 | 1.3x |
| 60% sparse | 5.2 | 1.3x | 21.8 | 1.5x |
| 70% sparse | 6.3 | 1.6x | 26.0 | 1.8x |
| 80% sparse | 8.0 | 2.0x | 31.9 | 2.3x |

**Table 20:** (SQuADv1.1/BERT-base) Speed-ups of pruned BERT-base models relative to the dense model, benchmarked with the sparsity-aware inference engine DeepSparse (version 1.0.2) (Kurtz et al., 2020; NeuralMagic, 2021) in two different scenarios on AMD EPYC 7702 64-Core Processor.

**Speed-ups of pruned BERT-base models.** In Table 20 we present speed-ups of our pruned models in the sparsity-aware CPU inference engine DeepSparse (Kurtz et al., 2020; NeuralMagic, 2021) (version 1.0.2). We consider two different scenarios and report speed-ups relative to the dense model benchmarked in the same environment.

**Robustness to one-shot pruning.** In Table 19 we present results for runs where CrAM is not used at every optimization step, and demonstrate that even in this setup the obtained models are still more robust to pruning compared to the models fully fine-tuned either with Adam or SAM optimizers reported in Table 4.

