# OpenReview forum: "CrAM: A Compression-Aware Minimizer"
_ICLR.cc/2023/Conference — ICLR 2023 poster_

### Official Review · Reviewer_fNyD · 2022-10-24

**Confidence:** 4
**Correctness:** 4
**Technical Novelty And Significance:** 3
**Empirical Novelty And Significance:** 3
**Recommendation:** 8

**Clarity, Quality, Novelty And Reproducibility:**

The paper is well-written and easy to follow. The idea of CrAM is somewhat novel and supported by theoretical analysis, but the experimental results are not significant compared with C-SAM. And the paper provides the details of algorithm and hyperparameters such that readers should be able to reproduce the main results.

**Strength And Weaknesses:**

Strength.
1) CrAM is simple to implement as part of a regular training loop and has a single scaling hyper-parameter.
2) CrAM enables training a dense model, which can later be compressed to different target levels, with minimal or no recalibration.
3) To derive the CrAM update, the paper provides the theoretical analysis to justify the choices made in designing the training method.
4) Sufficient Experiments demonstrate the wide range of application, including one-shot pruning, quantization, semi-structured N:M sparsity patterns and transfer learning.

Weaknesses.
1) For most of the comparisons, the paper choose the SAM as the target, such as Table 2,3,4 etc, which is not suitable because it trains models that potentially converge to flatter minima for better generalization rather than compressible models. It is more appropriate to compare with compression-aware C-SAM.
2) Compared with the gradual method DPF, CrAM performs worse at high sparsity, such as 90% and 95%. The paper explains that CrAM was not explicitly trained for such high sparsity. However, in the table 7, compared with C-SAM, CrAM still not perform very well at 80% and 90% sparsity with the same experimental configurations. It is better to clarify more about the phenomenon.
3) In the paper, the computation cost is not specific, such as twice as many forward-backward passes or the cost of BNT is minimal. It is better to show the cost explicitly in the comparison tables, such as the training time for single iteration.

**Summary Of The Paper:**

The paper proposes a new compression-aware minimizer dubbed CrAM that modifies the optimization step in a principled way, which results in dense models trained via CrAM can be compressible post-training, in a single step, without significant accuracy loss. Experimental analysis on image classification and language modelling shows that CrAM models can be pruned one-shot at a wide range of sparsity levels, while resulting in sparse models that are competitive with existing gradual pruning methods. Also, CrAM produces sparse models which perform well for transfer learning and works for semi-structured pruning patterns.

**Summary Of The Review:**

The idea of CrAM is overall novel and supported by theoretical analysis. And it is the first carry over the sharpness-aware idea to the task of obtaining compressible models. Although the experimental results are not significant compared with the closed works, the paper is likely to have modest impact within the subfield. Hence, I recommend for acceptance.

---

> ### Author Response · Authors · 2022-11-11
> **Response to Reviewer fNyD - Comparison with SAM or C-SAM**
>
> We thank the reviewer for the constructive comments and we will address the points made:
>
> 1. Comparing CrAM with SAM or C-SAM
>
> > It is more appropriate to compare with compression-aware C-SAM.
>
> We emphasize that C-SAM is an alternative update that we derived ourselves by following a similar approach to the one used in the original SAM paper. From our comparison in Appendix C.1 between CrAM and C-SAM, we conclude that while C-SAM is also robust to pruning, it can deteriorate the accuracy of the dense model (e.g. the comparison between CrAM-k70 and C-SAM-k70). Moreover, with CrAM we can explicitly optimize the dense model for free; this substantially improves the quality of the resulting model (please see Appendix Table 9). For these reasons, we focused on CrAM for our main experimental analysis.
>
> > For most of the comparisons, the paper choose the SAM as the target, such as Table 2,3,4 etc, which is not suitable
>
> Our goal, however, in the experiments from Tables 2 and 3 is to show that CrAM improves the robustness to compression even when it is used for a fraction of the training iterations; for this reason, we first need to determine a few baselines for the inherent “compressibility” of a model, and we choose SAM under the same setting, since it also requires two forward-backward passes at each optimization step. We further show on language models that CrAM improves the robustness to pruning even when it is used on a fraction of the iterations (please see Appendix Table 18). The purpose of Table 4, however,  is to compare different parameter settings for CrAM (e.g. training with different sparsities), which justify CrAM+Multi as our main method, used in the more exhaustive comparison from Table 5. We would like to point out that for both image classification and language modeling, we also compare CrAM-pruned models with existing pruning methods.
>
>
> 2. Performance of CrAM at high sparsity
>
> > Compared with the gradual method DPF, CrAM performs worse at high sparsity, such as 90% and 95%
>
> We would first like to emphasize that CrAM at 80% and 90% sparsity outperforms DPF on ImageNet/ResNet50 (please see Table 1).
>
>
> > compared with C-SAM, CrAM still not perform very well at 80% and 90% sparsity with the same experimental configurations
>
> Regarding C-SAM, we describe in Appendix C.1 that it can result in better sparse models at high sparsity. However, the differences in the Multi version of CrAM or C-SAM are negligible, considering the variance across runs. For CrAM/C-SAM with k{50,70}, indeed C-SAM 80% and 90% sparse models are substantially better, but they are still far below the accepted accuracy for these sparsity levels. As mentioned in the previous point, C-SAM can deteriorate the dense model accuracy, whereas with CrAM we can obtain an improvement (e.g. through the CrAM+ version). Notably, including the optimization of the dense model inside C-SAM would require an additional forward-backward pass. Therefore, we choose CrAM for our main experiments, as it achieves a better tradeoff between the quality of the dense model and slight improvements after one-shot pruning.
>
> 3. Computational cost of CrAM
>
> In the beginning of Section 4 we mention that on all our image classification experiments we use BNT on 1000 train samples, by performing 100 inference steps using batch size 128, with standard random augmentations. In total, this corresponds to performing inference on 12800 samples, which is a very minor fraction of the cost for validating a model on ImageNet. In our experiments, each CrAM step requires 2 forward-backward passes, followed by applying the Top-K operator. This implies a slightly higher training time than SAM, but we note that there exist efficient algorithms for Top-K which would make the cost of applying the operator negligible (for example K-th order statistic algorithm which runs in linear time). Furthermore, to address your comment, we explored whether the masks used in CrAM can be changed less frequently; we found out that updating the masks every 100th iteration did not change the results significantly. These experiments were performed on CIFAR10, and also in the CrAM-finetuning setup on ImageNet. We also note that with hardware support for sparse training, the total training cost of CrAM can be substantially reduced; such hardware support is already present (e.g. IPUs). We will include in the next revision a more detailed description regarding the specific costs of training with CrAM.

---

> > ### Comment · Reviewer_fNyD · 2022-12-10
> > **I am satisfied with the authors' answers**
> >
> > I am satisfied with the authors' answers. I will keep my score as "8: accept, good paper".

---

### Official Review · Reviewer_FqjP · 2022-10-26

**Confidence:** 4
**Correctness:** 3
**Technical Novelty And Significance:** 3
**Empirical Novelty And Significance:** 3
**Recommendation:** 6

**Clarity, Quality, Novelty And Reproducibility:**

The quality of this paper is good. Because the motivation, the addressed problem, the derivation of the solution, and the experiments are clear and easy to follow. The novelty of this paper is okay, as it proves the convergence of updating for their proposed CrAM.

**Strength And Weaknesses:**

Pros:
1. The paper is well-written and well-organized. It is easy to follow and read.
2. It is inspiring to leverage the flat minima for the model pruning task. This paper encourages the application of sharpness-aware training.
3. The methodology and appendix E are good for readers. The derivation is sufficient; the theoretical proof makes the paper complete.
Cons:
1. It would be better if the authors could have more experiments—for example, some parameter studies; some model pruning baselines. (One should be enough for the rebuttal period)
2. Appendix E is important; it would be better not to be placed the last part of the appendix.
3. There are too many variants of CrAM in the experiments. It is better to give a universal parameters set of CrAM.

**Summary Of The Paper:**

This paper aims to have a compression-aware minimizer that can compress dense DNN into a sparse one, and most importantly, without much performance drop. Inspired by the sharpness-aware minimizer, the authors leverage the flat minima to find the weights' coordinates that are stable to the perturbation. With the price of doubled computational overhead, the proposed compression-aware minimizer can prune dense models in a one-shot to 70-80% sparsity with a small drop in accuracy. The authors also give theoretical proof to guarantee the convergence of the proposed CrAM. The experiments on the ImageNet dataset verify the effectiveness of the proposed CrAM. The paper is well-organized and easy to follow.


**Summary Of The Review:**


To conclude, this paper is a good research paper. It is well-written, and the proposed method is sufficient in derivation and effectiveness. It would be better if some more experiments could be added.

---

> ### Author Response · Authors · 2022-11-11
> **Response to Reviewer FqjP**
>
> We thank the reviewer for the constructive comments! We address the following points made:
>
> 1. Parameter studies and model pruning baselines
>
> > It would be better if the authors could have more experiments—for example, some parameter studies
>
> Regarding parameter studies, we investigate the importance of different CrAM components (for example the extra gradient step), as well as alternative definitions, in Appendix C.1. For each comparison, we perform a new grid search over the $\rho$ hyperparameter, where this is present. Results clearly show that the extra gradient step in CrAM improves the overall robustness to one-shot pruning, at all sparsity levels. Furthermore, with our alternative definitions (e.g. C-SAM) we can obtain more prunable models at high sparsity (please see Table 7), but at the price of reduced dense model accuracy. This motivates our choice of CrAM over other definitions, as we can also explicitly train the dense model at no extra cost (through CrAM+), while also obtaining very competitive sparse models. We also further explore the impact of CrAM+ and of using sparse gradients in the final update in Appendix C.2.
>
> > model pruning baselines
>
> We would like to emphasize that we compare our results after one-shot pruning with those obtained with existing state-of-the-art pruning methods (please see for example Tables 1 and 5). For ImageNet/ResNet50, our 80% and 90% one-shot pruned models outperform existing pruning methods STR and DPF (Table 1). Moreover, at 90% sparsity, we are still competitive with state-of-the-art pruning methods AC/DC and WoodFisher (notably, WoodFisher requires a pre trained dense model). We also compare CrAM with existing pruning methods on language modeling (Table 5), where we show that at 70% sparsity CrAM matches or outperforms all pruning methods considered, including gradual ones. Furthermore, at 80% sparsity we obtain a competitive model that outperforms all methods, except PLATON, which performs gradual pruning. With additional sparse finetuning of a few epochs we outperform by a wide margin all pruning methods considered (even though they are gradual, and perform independent runs for each sparsity target, whereas we perform one-shot pruning).
>
> 2. Importance of Appendix E
>
> Thank you for the suggestion! We agree that the theoretical justification is important, and we will rearrange the content to give it more priority.
>
> 3. Multitude of CrAM variants
>
> We acknowledge that there are multiple versions of CrAM discussed, and we will highlight the CrAM+ Multi version as our method of choice. In fact, with CrAM+ Multi we obtain our best results for one-shot pruning on ImageNet and SQuAD, as well as for sparse finetuning. We will add more clarifications on this point in the next major revision.

---

### Official Review · Reviewer_VFwD · 2022-10-28

**Confidence:** 4
**Correctness:** 2
**Technical Novelty And Significance:** 2
**Empirical Novelty And Significance:** 2
**Recommendation:** 6

**Clarity, Quality, Novelty And Reproducibility:**

* The paper directly borrows the SAM framework and tests a few different algorithmic changes in the order of compression operations. In that sense it is hard to say the idea is original.
* In my view writing should improve in terms of organization, focus, and clarity.


**Strength And Weaknesses:**

Strength
* provide a range of experiments making comparisons on different scenarios, baselines, and domains
* develop theory based on the mini max framework, by which compression interpreted as perturbation can be effectively performed as robust optimisation
* N:M sparsity pattern is nice in practice adding it is definitely plus

Weakness
* CrAM, the basic version of the method, does not work well, whereas the provided theoretical justification is supposed to indicate CrAM should be enough, not CrAM+. The theoretical justification itself is only weakly justifying the method. For instance treating the mask operator and straight-through estimator the way the author does is very weakly justified.
* overclaim for generality of the method. present as if it works for any compression methods, but mostly focuses on top-k based on magnitude of parameters. a quantization scheme is provided in the appendix but not enough to claim for compression in general.
* far below the sparsity level compared to existing methods. present some effectiveness up to 90%, but considering the sparsity literature it is far below. for 95% sparsity it is a level at which the method does not work.
* coated with various experiments, yet fails to concentrate on the central part. Although they conduct many interesting experiments, they are often quite partial point-by-point based and far from being comprehensive.
* The only effective result is that CrAM/CrAM+ can give a network that is one-shot prunable. However, this is not considered enough given the fact that this method requires additional hyperparameter tuning such as the choice of sparsity levels during training, that there exist other one-shot training (and pruning afterwards) methods without requiring SAM framework or double the forward-backward computations.


**Summary Of The Paper:**

This paper presents a method to train a neural network, CrAM (and CrAM+), which results in a trained network that can be compressed (by pruning or quantization) while maintaining accuracy. The method is developed based on plugging compression in the SAM’s inner maximization step; the perturbed parameters are obtained by taking the ascent step (for maximization) and applying some compression on them. The authors attempt to provide theoretical justification for such an algorithm based on robust optimization literature. The authors present details on some extensions including batch norm tuning or SAM+multi, and further, experimental results on Imagenet as well as SQuAD for different comparisons.


**Summary Of The Review:**

The paper explores an interesting combination of SAM and sparsity, but importantly lots of parts in the paper is not quite tight and needs further exploration.

---

> ### Author Response · Authors · 2022-11-11
> **Response to Reviewer VFwD - Generality and goals of our method**
>
> We thank the reviewer for the feedback and we will answer point-by-point all the concerns raised.
>
> 1. Theoretical justification of CrAM and CrAM+
>
> > the provided theoretical justification is supposed to indicate CrAM should be enough, not CrAM+.
>
> We very respectfully disagree with this statement: in particular, the CrAM+ update is also justified theoretically, just as basic CrAM.
> For this, please recall that our theoretical analysis proves that the CrAM update we propose gives a descent direction for the CrAM loss function. CrAM+ is simply the sum of the CrAM loss and the standard cross entropy loss. Since the gradient clearly gives a descent direction, together with our previous result on CrAM, we immediately obtain a descent direction for CrAM+.
>
> > The theoretical justification itself is only weakly justifying the method. For instance treating the mask operator and straight-through estimator the way the author does is very weakly justified.
>
> First, notice that randomly sampling sparsity levels during training is theoretically-justified as well, as we are simply optimizing the expected CrAM loss where the expectation is taken over *compression levels*: at each step, we estimate the mean over sparsity levels using a random sample.
>
> Regarding our use of the straight-through estimator, we note that we are using standard techniques used across all work on pruning and quantization (e.g., DPF, QAT), so a similar criticism can be made to the entire area.
>
> Specifically, our approach of using sparse gradients is motivated by the observation that the mask changes very little during training; we justify this empirically in Appendix C.2. Furthermore, the exact same type of straight through gradient estimator is also widely in practice (for example the gradient for the ReLU function), and it is well-known that using the identity function can be suboptimal [1]. We provide more explanations on the impact of sparse gradients in Section 3.3 and Appendix C.2.
>
> 2. Generality of the method
>
> According to [Hoefler et al., JMLR2021], there are essentially four types of model compression: unstructured,  semi-structured (2:4 sparsity), structured sparsity, and quantization.
>
> Our experimental study shows that CrAM+ results in compressible accurate models on 3 of these 4 types: unstructured (at multiple sparsity levels, both global and uniform), semi-structured (2:4 and 4:8 patterns, supported in hardware) and quantization (4 bits), on two different applications: CNN image classification on ImageNet and CIFAR, and Transformer language modeling tasks. We thus believe this justifies our claim of generality for the method.
>
> For comparison, prior work in the same setting [Miao et al., ICLR 2022] only considered a single compression type (unstructured sparsity), on smaller image classification tasks (CIFAR and TinyImagenet).
>
> [1] Yin, Penghang, et al. "Understanding straight-through estimator in training activation quantized neural nets." ICLR 2019.
>
> [2] Miao, Lu, et al. "Learning Pruning-Friendly Networks via Frank-Wolfe: One-Shot, Any-Sparsity, And No Retraining." ICLR 2022
>
> [3] Hoefler, Torsten, et al. "Sparsity in Deep Learning: Pruning and growth for efficient inference and training in neural networks." J. Mach. Learn. Res. 22.241 (2021): 1-124.

---

> > ### Author Response · Authors · 2022-11-11
> > **Response to Reviewer VFwD - Part 2**
> >
> >
> > 3 . Performance at higher sparsities
> >
> > > far below the sparsity level compared to existing methods. present some effectiveness up to 90%, but considering the sparsity literature it is far below.
> >
> > Regarding performance relative to sparsity, as can be seen in Table 1, our method outperforms DPF on ResNet50/ImageNet at both 80% and 90% sparsities, and is competitive with much more complex and expensive second-order and regularization methods. We emphasize that this is in an unfair setting for our method: all other methods are re-run for each sparsity in turn, whereas CrAM works simply in one-shot after one run, and produces an accurate dense model as well.
> > As can be seen in Table 5, our method outperforms several other non-trivial gradual baselines from the literature on language modeling as well.
> >
> > > for 95% sparsity it is a level at which the method does not work.
> >
> > We believe that the reviewer is claiming that our method “does not work” based on a single datapoint: 95% sparsity results on CIFAR-10/ResNet20, for Cram-Multi+[30-90].
> >
> > Please first note that our method outperforms both DPF and the one-shot method SFW at the same 95% sparsity on the VGG-16 network, even though DPF is executed independently for each sparsity. The results are located in the same table (Table 6).
> >
> > The experiments on ResNet18 discussed in the text below the table clearly show that our method works at sparsity *much higher* than 95%, and significantly improves upon comparable one-shot methods:
> > On ResNet18 we maintain the model accuracy after pruning one-shot at 96% sparsity (95.4%, after BNT) and have a 1.3% drop at 98% sparsity (94.4% Top-1). This improves significantly on Miao et al. (2022) and Zimmer et al. (2022), which both obtain ≤ 93% accuracy at 95% sparsity, in exactly the same setting.
> >
> > To address the precise CIFAR-10/ResNet20 result, we have simply re-run our method with the sparsity levels [50-95%]: specifically, notice that the 95% was not previously included among the sampled levels. We stress that this is a fair comparison, as DPF does individual runs *for every sparsity level.*
> > In this case, we outperform DPF at both 90% sparsity (CrAM: 91.15%; DPF: 90.9%) and at 95% sparsity (CrAM: 89.2% , DPF: 88%).  Our results are averaged across three runs from different seeds.
> >
> > In sum, these results show that we outperform all other similar previous methods, even at high sparsity, on both large-scale (ImageNet, SQuAD) and small-scale (CIFAR-10) tasks.
> >
> > We thank the reviewer for the opportunity to clarify this point, and are happy to engage in further discussion.
> >
> >
> > 4.  Paper focus
> >
> > We reiterate our goal: we want a method that can train models which not only have good accuracy, but can also be easily compressible post-training, in a single step, without retraining. Our work is centered around answering the following questions, which support this central goal:
> >
> > **Q1**: Does CrAM/CrAM+ result in high quality dense models, which can be easily pruned, compared to training using standard methods?
> > **A1**: Yes, for standard image classification (Figure 1, ImageNet) and language modeling (Table 4, SQuAD).
> >
> > **Q2**: Are one-shot pruned CrAM models competitive with sparse models obtained through gradual pruning methods?
> > **A2**: Yes, for both image classification - Table 1 - and language modeling - Table 5, even with substantial improvements after short fine-tuning, outperforming specialized methods.
> >
> > **Q3**: Is CrAM competitive with alternative one-shot methods, such as Frank-Wolfe-inspired approaches?
> > **A3**: Yes, the results in Section 4.3 show that we significantly outperform these methods.
> >
> > **Q4**: Since CrAM is computationally more expensive, can it be used for short finetuning or on fewer training iterations?
> > **A4**: Yes: see finetuning experiments in Tables 2 and 3 for image classification, and results with infrequent CrAM updates for language models in Appendix Table 18.
> >
> > **Q5**: Does CrAM support other types of compression?
> > **A5**: Yes, it also supports semi-structured sparsity and quantization.
> >
> > We do acknowledge the reviewer’s comments on presentation, and in particular on clarifying our main method and its theoretical guarantees. We will implement them in the next major revision of our work.
> >
> > At the same time, we feel that we should not be penalized for providing thorough comparisons between alternative updates (C-SAM, CrAM, CrAM-Multi, CrAM+) in order to identify the key aspects of the method, and how they interact.

---

> > > ### Author Response · Authors · 2022-11-11
> > > **Response to Reviewer VFwD - Part 3**
> > >
> > >
> > > 5.   Hyperparameters and comparison with other methods
> > >
> > > As the reviewer points out, with CrAM we can obtain a model that is one-shot prunable. Further, we also obtain a dense model with higher accuracy than regular SGD training.
> > >
> > > Furthermore, our method uses minimal hyperparameter tuning: we literally used the same training setup (learning rate, momentum value, weight decay) as for standard SGD training, and on ImageNet we have kept the same value for the extra gradient step as the one used in the original SAM paper ($\rho$=0.05).
> > > It is true that we sample multiple sparsity levels during training, but these values are quite intuitive: we simply choose a range of sparsities including our sparsity targets, which makes sense given that we want to be robust against multiple levels post-training (e.g. for CrAM+ Multi on ImageNet we used 50%, 70% and 90%).
> > >
> > > Regarding the comparison with SFW methods, we would first like to point out that these methods provide experiments on smaller datasets (e.g. CIFAR10/TinyImageNet) and using highly-overparameterized architectures (e.g. VGG-16 and ResNet18).
> > >
> > > We are the first to show results on large scale tasks, such as ImageNet/ResNet50 and SQuAD using BERT.  Moreover, in the same settings as the SFW, we obtain better results for the dense model and also after one-shot pruning, even at high sparsity (we further explain this in point 3 above).
> > > Furthermore, we emphasize that SFW methods, particularly [Miao et al, ICLR 2022], require in fact substantially more hyperparameter tuning than CrAM: in addition to the sparsity level used during training, also the initial learning rate, the non-standard learning rate scheme and the radius of the K-sparse polytope play a very important role in the success of the method (please see Figure 1c, or Figure 6 in [Miao et al, ICLR 2022]).
> > >
> > > It is true that SFW methods do not require an additional forward-backward pass per step, but we point out that with hardware specialized to handle sparse training, e.g. IPUs, the computational complexity of the second sparse forward-backward pass can be reduced substantially. Moreover, we have shown that our method is effective in post-training fine-tuning mode as well.

---

### Official Review · Reviewer_6AwX · 2022-10-31

**Confidence:** 3
**Correctness:** 3
**Technical Novelty And Significance:** 2
**Empirical Novelty And Significance:** 3
**Recommendation:** 6

**Clarity, Quality, Novelty And Reproducibility:**

This paper is well written and easy to understand.

The derivation and formulation in this paper are in good quality.

The novelty of this paper is not significant, as it's very close to once-for-all and other supernet based methods in essence. (see "Strength And Weaknesses" for more details.)

**Strength And Weaknesses:**

Strength
- The proposed method can train a single set of model weights which performs well on multiple compression rate, i.e. sparsity ratios.
- The formulation is a good extension of sharpness-aware optimization to the direction of optimization with constraints.
- The authors also considered and discussed the computation complexity of the algorithm.
- The experiments result show that the dense model results can be even better compared to training a standalone dense model.

Weaknesses
- Although the paper derives the method from the view of optimizer, the updating rule is almost the same as using SAM optimizer on supernet/one-shot NAS methods such as once-for-all network[1], bignas[2]. The novelty of this method is not very significant from this point of view.
- It's better to have some discussion and comparison to the method which directly combines SAM with once-for-all network, e.g., is there any difference in the actual updating rule and how much difference in the results.

[1] Cai, Han, et al. "Once-for-all: Train one network and specialize it for efficient deployment." arXiv preprint arXiv:1908.09791 (2019).

[2] Yu, Jiahui, et al. "Bignas: Scaling up neural architecture search with big single-stage models." European Conference on Computer Vision. Springer, Cham, 2020.

**Summary Of The Paper:**

This paper proposed a specialized optimizer for optimizing the DNN to have good compression ability, i.e. the optimized model can be pruned/quantized to smaller models which have good accuracy.

The paper is based on a formulation related to the sharpness aware minimization, by adding a compression related constraint projection on the inner "sharpness-aware" gradient ascent step.

Experiments result show that the proposed method can train a model which not only performs good on dense model but also on sparse models.

**Summary Of The Review:**

The paper provides an intact derivation of extending SAM optimizer to the context of constrained optimization and especially model compression.

The experiments validate the proposed algorithm on different sparsity ratios and also dense model.

From the updating rule, the proposed algorithm is the almost the same as using SAM on once-for-all network, so the novelty of this paper is not very significant.

---

> ### Author Response · Authors · 2022-11-11
> **Response to Reviewer 6AwX - Relationship between CrAM and OFA/BigNAS**
>
> We thank the reviewer for the comments and suggestions!
> We would like to discuss the points made by addressing two important aspects:
>
> 1. CrAM is complementary to OFA/BigNAS in terms of objective, training setting and costs
>
> Specifically, OFA/BigNAS aim to train a model that can be “mapped” to different architecture settings (e.g. widths, depths, kernel sizes) to allow for easy deployment, with the primary application being inference on mobile devices. Thus, OFA/BigNAS focus on *structured sparsity*, by changing the architecture in very specific ways. By contrast, CrAM can be adapted to work with different types of compression: we show results for unstructured, semi-structured sparsity and quantization, which are efficiently supported on CPU and GPU architectures. So the methods are complementary in terms of scope and applications.
>
> Second, we note that OFA has high training costs, as it requires
>
> a. Training the full model;
>
> b. Searching through the space of possible subnetworks, by  progressively shrinking the model size in several stages, with each stage separated by a long finetuning period;
>
> c. Further, as training progresses, each shrinking stage requires more forward-backward passes through the model (e.g. at the final stage, four different sub-networks are sampled at each step, and their gradients are aggregated for the final model update);
>
> d. Lastly, the authors report their best results after additionally fine tuning for 75 epochs the specialized sub-networks.
>
> BigNAS has a similar goal and approach to OFA; however, at every optimization step BigNAS does four forward-backward passes through the sampled models, following the approach from [Yu & Huang], which we discuss in our Related Work section. We emphasize that this should not be seen as a criticism w.r.t. OFA/BigNAS, as careful training, sampling, and finetuning may in fact be *necessary* to provide accurate *structured* compression of models.
>
> By contrast, CrAM uses the standard number of epochs for training our CrAM models, and minimal hyperparameter tuning. We only sample one compressed model per step, and our final model improves the accuracy over the baseline. Our one-shot pruned models require no additional finetuning to be competitive with gradual pruning methods on both ImageNet/ResNet50 and BERT-base models, as shown in Tables 1 and 5.
>
> Thus, in our view, the methods are also complementary in training setting and costs, with CrAM providing a cost-efficient way of obtaining highly-accurate compressed and dense models for CPU and GPU deployments.
>
> 2. The novelty of CrAM, compared to using SAM with OFA or BigNAS
>
> To address this, please observe that directly using SAM on a compressed model would correspond to optimizing the following loss function:
>
> $$\max_\epsilon L(C(\theta) + \epsilon).$$
>
> Following the SAM derivations to solve this maximization problem would result in perturbing the *compressed* model using its gradient.
>
> By contrast, please note that the CrAM loss function is different, i.e. $\max_\epsilon L(C(\theta+\epsilon))$.
>
> Specifically, we perturb the *dense* model using its gradient, followed by compression of this perturbation. Importantly, with our formulation we can also optimize the dense model *for free*, as we already have access to its gradient.
>
> We agree that CrAM can be used for training OFA and BigNAS at no extra cost, since these methods already compute the gradient of the dense model at each optimization step. Yet, simply using SAM would *double* computational costs of training OFA and BigNAS, which, as already discussed in part (1) of our answer, are already very high.
>
> We thank you for this interesting suggestion, which we plan to investigate in future work. We will also add a clarification about the differences between CrAM and using SAM on sparse models.
>
>
> We would be happy to continue the conversation if there are any further questions!
>
> [Yu & Huang] Yu, J., Huang, T.: Universally slimmable networks and improved training techniques. ICCV 2019

---

### Author Response · Authors · 2022-11-17
**Discussion Reminder and Response Summary**

Dear Reviewers,

Thank you again for the feedback on our submission!

We would like to kindly remind you about our responses, which provide detailed explanations for all the points raised.  Specifically, we address the following:
- comparison between CrAM and OFA/BigNAS methods
- theoretical justification of CrAM and CrAM+
- generality of our method
- performance at high sparsity and comparison with gradual pruning methods; we further provide a revision containing improved one-shot pruning numbers for CrAM+Multi on CIFAR10/ResNet20, which show that CrAM can surpass DPF even at 95% sparsity
- comparing CrAM with SAM on the fine-tuning tasks

We also acknowledge and will address the suggestions regarding the presentation of our method in a future revision. However, for now we believe it is better for the clarity of the discussion to keep the current format of the sections. We will also include in the final revision more details on the training cost of CrAM and a discussion on the differences between CrAM and OFA/BigNAS.

We are looking forward to engaging in further discussion about our submission!

Best regards,

The authors

---

### Comment · Area_Chair_xq1n · 2022-11-20
**Please update your reviews**

Please make sure that your reviews acknowledge authors’ responses and reflect your current evaluation of the paper. This is particularly important if you didn’t directly engage with the authors during the discussion phase (so the authors don’t know if their response changed your evaluation) or if you expressed an intention to update your rating but did not do so.

Cheers,
AC

---

### Decision · Program_Chairs · 2023-01-20

**Decision:**

Accept: poster

**Justification For Why Not Higher Score:**

To some extent, the proposed method is a good extension of the SAM on the network compression task. The performance is good, and the derivation is reasonable. However, this paper is not inspiring enough to the community. The spotlight paper of ICLR needs more novelty and exploration.

**Justification For Why Not Lower Score:**

As all the reviewers gave acceptance scores to the paper, the proposed method is practical in the compression task. There are a few good extensions of SAM work; this paper is a good extension of SAM. Therefore, this paper could be published in ICLR.

**Metareview: Summary, Strengths And Weaknesses:**

This paper presents a compression-aware minimizer (CrAM and CrAM+) to compress the dense Deep Neural Networks (DNN) with preserved performance. The presented method is inspired by the inner maximization step (parameters perturbation) of Sharpness-aware minimization (SAM) to filter out the stable parameters. Experimental analysis well supports the efficacy of CrAM and CrAM+.

Strength:
-Reasonable method design and clear writing
-A good formulation and extension of SAM.
-Clear writing and extensive experiments.

Weakness:
-Need more discussion about the sparsity level and generalization of CrAM.
-The additional computational overhead of CrAM (2X).


**Note From Pc:**

if the above contains the word "oral" or "spotlight" please see: "oral" presentation means -> notable-top-5% and "spotlight" means -> notable-top-25%. As stated in our emails, we are disassociating presentation type from AC recommendations

**Summary Of Ac-Reviewer Meeting:**

N/A